# Apparent contradiction in the projected climatic water balance for Austria: wetter condition on average versus higher probability of meteorological droughts.

Klaus Haslinger[1,*], Wolfgang Schöner[2], Jakob Abermann[2], Gregor Laaha[3], Konrad Andre[1], Marc Olefs[1], Roland Koch[1],

[1]GeoSphere Austria, Climate-Impact-Research, Hohe Warte 38, 1190 Vienna, Austria
[2]University of Graz, Department of Geography and Regional Sciences, Heinrichstraße 36, 8010 Graz, Austria
[3]University of Natural Resources and Life Sciences (BOKU), Peter Jordan Straße 82, 1190 Vienna, Austria
*Correspondence to*: Klaus Haslinger (klaus.haslinger@geosphere.at)

**Abstract.** In this paper future changes of surface water availability in Austria are investigated. We use an ensemble of downscaled and bias-corrected regional climate model simulations of the EURO-CORDEX initiative under moderate mitigation (RCP4.5) and Paris agreement (RCP2.6) emission scenarios. The climatic water balance and its components (rainfall, snow melt, glacier melt and atmospheric evaporative demand) are used as indicators for surface water availability and we focus on different altitudinal classes (lowland, mountainous and high alpine) to depict a variety of processes in complex terrain. Apart from analysing the mean changes of these components we also pursue a hazard risk approach by estimating future changes in return periods of meteorological drought events of a given magnitude as observed in the reference period. The results show in general wetter conditions over the course of the 21st century over Austria on an annual basis compared to the reference period 1981-2010 (e.g. RCP4.5 +107 mm, RCP2.6 +63 mm for the period 2071-2100). Considering seasonal differences, winter and spring are getting wetter due to an increase in precipitation and a higher fraction of rainfall as a consequence of rising temperatures. In summer only little changes in the mean of the climatic water balance conditions are visible across the model ensemble (e.g. RCP4.5 ±0mm, RCP2.6 -2 mm for the period 2071-2100). On the contrary, by analysing changes in return periods of drought events, an increasing risk of moderate and extreme drought events during summer is apparent, a signal emerging within the climate system along increasing warming.

**Short Summary:**
Future changes of surface water availability in Austria are investigated. Alterations of the climatic water balance and its components (liquid precipitation, snow melt, potential evapotranspiration) are analysed along different levels of elevation. Results indicate in general wetter conditions with particular shifts in timing of the snow melt season. On the contrary, an increasing risk for summer droughts is apparent due to increasing year-to-year variability and decreasing snow melt under future climate conditions.

# 1 Introduction

Drought and water scarcity are among the most devastating natural hazards causing damage on various natural and human systems. Average annual economic losses from drought alone are estimated to 9 billion Euros in the European Union (European Commission, 2020). Europe was struck several times in recent years by severe summer droughts causing enormous economic damage, for example the drought of 2015 (Laaha et al., 2017; Van Lanen et al., 2016; Ionita et al., 2017) and of 2018 (Buras et al., 2020a; Boergens et al., 2020; Bakke et al., 2020) which hit Austria in particular. Future climate change will further alter

hydroclimatological conditions in various ways through e.g. shifts in rainfall distribution through intensification of the hydrological cycle (Allan et al., 2020; Vargas Godoy and Markonis, 2022), shifts in seasonality of certain variables (e.g. snow, Mudryk et al., 2020) and large scale changes in the atmospheric circulation and moisture transport (Fabiano et al., 2021). It is therefore vital to assess possible future changes of multiple input, output and storage terms at the land surface in order to unravel critical processes and thresholds in both space and time which may impact surface water availability.

Austria with its mountainous topography is in general considered as a water-rich country with freshwater resources by far exceeding the demand (Haas and Birk, 2019; Stelzl et al., 2021). Recent drought years, however, raised concerns about changing water availability. Precipitation trends on the very long term back to the 19$^{th}$ century show no significant trend and changes are mostly subject to multidecadal variability (Brunetti et al., 2009; Haslinger et al., 2021). During the past decades precipitation slightly increased, though this signal did not appear in the runoff signatures, since it was balanced by increasing

atmospheric evaporative demand (Duethmann and Blöschl, 2018). Precipitation in the form of snow plays an important role for surface water availability in mountainous areas. In Austria and the Alpine region in general a significant decline in snow depth is observed (Matiu et al., 2021; Olefs et al., 2020; Schöner et al., 2018) with possible impacts on consequent summer low flows (Jenicek et al., 2016). Considering drought conditions in particular, meteorological droughts show no trends over the past 200 years (Haslinger et al., 2019b; Haslinger and Blöschl, 2017). On the contrary, hydrological droughts exhibit

negative trends over the past 40 years, but only over some lowland areas in the North and Southeast of Austria (Laaha et al., 2015; Blöschl et al., 2018).

Climate change already alters some aspects of water availability in Austria, mainly due to decreasing snow cover and increasing atmospheric evaporative demand. Climate projections show an increase in precipitation over Austria which is stronger in winter and spring compared to summer (Blöschl et al., 2018). Increasing temperatures also act on the future snow cover with

specific impacts on drought development and predictability (Livneh and Badger, 2020; Musselman et al., 2021). For Austria in particular, Olefs et al. (2021) highlighted the sensitivity of snow cover to temperature especially below 1500 m a.s.l. Future scenarios for meteorological drought conditions show increasing drought risk particularly during summer under IPCC CMIP3 scenarios (Haslinger et al., 2016; Laaha et al., 2015) which is mainly driven by precipitation decrease and atmospheric evaporative demand increase (see also Gali Reniu, 2017). For river discharge IPCC CMIP3 projections point towards

decreasing summer low flows in lowland areas and increasing winter low flows in the alpine areas of Austria (Laaha et al., 2015; Parajka et al., 2016). Although the body of existing literature points towards changing water availability in Austria a

comprehensive synopsis of all relevant processes altering surface water availability is not accomplished yet, or just for small spatial entities (Hanus et al., 2021). Here we aim to fill this research gap by addressing following research questions using the Austrian reference climate scenario dataset based on EURO-CORDEX CMIP5 regional climate simulations:

(i) How will future surface climatic water balance change under different emission scenarios and different elevations?

(ii) How do the individual components of the surface water balance change during the course of the year?

(iii) How will the probability for extreme drought conditions change under future climate?

## 2 Data

In this study we use gridded observations and modelled data. All datasets are on a congruent, 1 km grid and fully cover the Austrian domain, see Figure 1a for the domain boundaries as indicated by coloured topography shading. Considering climate scenarios we use the Austrian national reference scenario dataset OEKS15 (Chimani et al., 2016) which consists of a selected ensemble of regional climate model (RCM) simulations driven by CMIP5 global climate models from the EURO-CORDEX EUR-11 database. The selection of the models is based on quantitative criteria as described in Chimani et al. (2020). Three

different emission scenarios are available within OEKS15, here we use RCP4.5 and RCP2.6. With this choice we intend to depict on one hand a likely outcome of emission pathways during the 21st century, where RCP4.5 draws a modest climate change mitigation future and a likely outcome, and on the other hand a more favourable outcome by meeting the Paris agreement within the scenario pathway of RCP2.6. The broadly used RCP8.5 scenario is intentionally not included here, since its emission pathway is highly unlikely from today´s emissions trajectories, as well as current and pledged policies and is often

misleadingly used as a business as usual scenario (Hausfather and Peters, 2020; Pielke and Ritchie, 2021). From today's perspective an emission path following RCP4.5 is, at least until 2030, the most likely one given current estimates (UNFCCC, 2022).

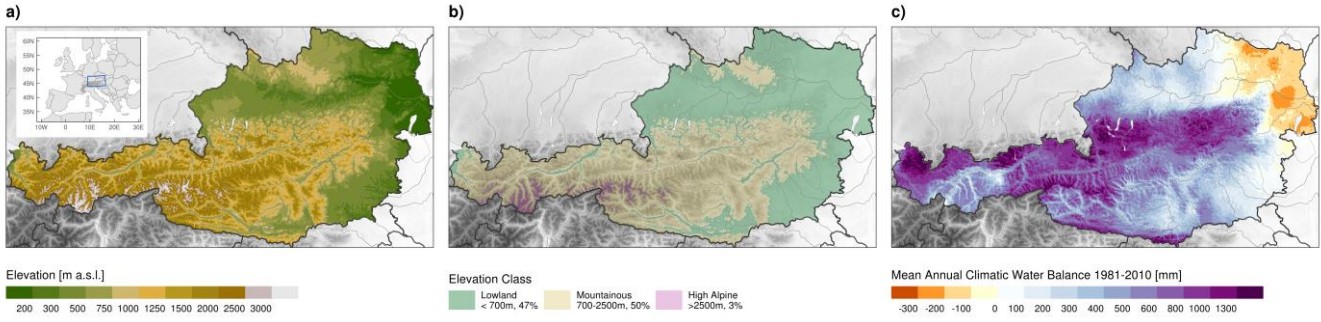

**Figure 1: (a) Topography of Austria, the inset figure indicates the location on the European domain, (b) altitudinal classification, (c)**
**long term mean (1981-2010) annual climatic water balance**

In total 16 model runs are available for RCP4.5 and 8 for RCP2.6, a summary is given in Table 1 indicating the driving global climate model, RCM and member realization. The EURO-CORDEX simulations are downscaled and bias-corrected using

scaled distribution mapping (Switanek et al., 2017) which is an optimized quantile mapping approach (Themeßl et al., 2011) preserving the initial climate change signal of the RCM simulation. As reference datasets for the bias correction, gridded observations of daily maximum and minimum temperatures of SPARTACUS (Spatiotemporal Reanalysis Dataset for Climate in Austria, Hiebl and Frei, 2016) are used as well as GPARD1 (Gridded Precipitation for Austria at Daily 1 km Resolution, Hofstätter et al., 2015) for daily precipitation sums. Both reference datasets are considering orographic effects on temperature (e.g. cold air pool formation, foehn effects) and on precipitation (orographic precipitation) which is rather important for interpolating climatic variables in complex terrain of the Austrian domain. The basic data sets for OEKS15 (EURO-CORDEX) were thoroughly evaluated in Kotlarski et al. (2014) and OEKS15 was evaluated and a comprehensive guide line given on the usage in Chimani et al. (2020).

To account for different processes considering changes in water availability along elevation we stratified the Austrian domain in three different classes of elevation (Figure 1b). The first denotes for the lowland areas below 700 m a.s.l. (47% of the entire domain), which are mostly comprised of agricultural land and also encompasses the major settlement areas and large urban areas. The second elevation class defines mountainous areas between 700 and 2500 m a.s.l. (50% of the entire domain). These are mostly covered by forests and alpine pastures. The third elevation class denotes for high alpine areas above 2500 m a.s.l. with some alpine pastures and mostly unvegetated terrain and glaciers at the highest altitudes (3% of the entire domain).

**Table 1: RCM simulations used in the present study**

| ID | Institute | Global Climate Model | Regional Climate Model | Member realization | RCP4.5 | RCP2.6 |
|----|-----------|----------------------|------------------------|--------------------|--------|--------|
| 1 | Météo France | CNRM-CM5 | CCLM4-8-17 | r1i1p1 | X | |
| 2 | Météo France | CNRM-CM5 | CNRM-ALADIN53 | r1i1p1 | X | |
| 3 | Météo France | CNRM-CM5 | SMHI-RCA4 | r1i1p1 | X | |
| 4 | Irish Centre for High-End Computing | EC-EARTH | CCLM4-8-17 | r12i1p1 | X | X |
| 5 | Irish Centre for High-End Computing | EC-EARTH | RACMO22e | r12i1p1 | | X |
| 6 | Irish Centre for High-End Computing | EC-EARTH | RCA4 | r12i1p1 | X | X |
| 7 | Irish Centre for High-End Computing | EC-EARTH | RACMO22e | r1i1p1 | X | |
| 8 | Irish Centre for High-End Computing | EC-EARTH | HIRHAM5 | r3i1p1 | X | X |
| 9 | Institut Pierre Simon Laplace | IPSL-CM5A-MR | WRF331f | r1i1p1 | X | |
| 10 | Institut Pierre Simon Laplace | IPSL-CM5A-MR | RCA4 | r1i1p1 | X | |
| 11 | Met Office Hadley Centre | HadGEM2-ES | CCLM4-8-17 | r1i1p1 | X | |
| 12 | Met Office Hadley Centre | HadGEM2-ES | RCA4 | r1i1p1 | X | X |
| 13 | Max Planck Institute for Meteorology | MPI-ESM-LR | CCLM4-8-17 | r1i1p1 | X | |
| 14 | Max Planck Institute for Meteorology | MPI-ESM-LR | REMO2009 | r1i1p1 | X | X |
| 15 | Max Planck Institute for Meteorology | MPI-ESM-LR | RCA4 | r1i1p1 | X | X |
| 16 | Max Planck Institute for Meteorology | MPI-ESM-LR | REMO2009 | r2i1p1 | X | X |
| 17 | Norwegian Climate Center | NorESM1-M | HIRHAM5 | r1i1p1 | X | |

## 3 Methods

### 3.1 The Climatic Water Balance

In this paper we use the climatic water balance (CWB) as the basic metric for assessing surface water availability and drought conditions. In principal, the CWB is the difference between precipitation and atmospheric evaporative demand (AED) and is therefore able to depict both atmospheric supply and demand. It is often used to derive the standardized precipitation evapotranspiration index (SPEI, Vicente-Serrano et al., 2010) by fitting a probability distribution function to the CWB and

afterwards transforming it into a unit normal distribution. This enables a rather intuitive assessment of the moisture conditions (negative/positive values dryer/wetter than normal) and has made the SPEI a broadly applied index for drought monitoring and forecasting but for research purposes as well (e.g. Haslinger et al., 2014, 2016). Here we stick to the basic CWB to be able to give absolute values of change rather changing index values which may be difficult to interpret. The annual mean CWB for the 1981-2010 period is displayed in Figure 1c. It shows a rather diverse spatial pattern, with positive values in the mountainous, western parts of Austria in contrast to distinct negative values in the flat, low elevation part in the northeast of the country. In general this pattern is mainly driven by spatial patterns of precipitation with largest precipitation amounts occurring in the so called Northern Stau regions and the decrease of AED along higher elevations.

For analysing the impacts of future climate change on CWB evolution we extended this concept by considering the effects of snow accumulation and ablation as well as phase conditions of precipitation (liquid versus solid). This enables to assess the changing snow cover conditions along projected temperature increases and potential shifts of water availability during the course of the year and across different elevation zones. Hence for this analysis CWB is given by the following equation:

$$CWB = (R + M) - AED$$

Where R stands for liquid precipitation or rainfall, M for snow melt and AED for atmospheric evaporative demand. For the special case of the High Alpine area (c.f. Figure 1b) we also consider glacier melt as an individual positive term in the climatic water balance equation.

### 3.2 Atmospheric Evaporative Demand

Atmospheric evaporative demand (AED), or reference evapotranspiration, refers to the maximum moisture flux to the atmosphere from a standardized land surface (grass) under continuous moisture supply and given meteorological conditions (Lhomme, 1997). It is therefore independent from soil properties, hence it is widely used to assess crop water requirements for example. In this study we use AED estimates following the approach of Haslinger and Bartsch (2016). The authors used the method of Hargreaves (Hargreaves and Allen, 2003; Hargreaves and Samani, 1985) which requires daily maximum and minimum air temperature and latitude as input data. The authors re-calibrated the original Hargreaves parameter against FAO-Penman-Monteith (Allen and Food and Agriculture Organization of the United Nations, 1998) estimates on several stations across Austria. This new parameter set was then interpolated in space and time (during the course of the year) which was then used along the other input dataset for calculating AED.

The final gridded AED product (ARET, Austrian Reference EvapoTranspiration dataset) was forced by daily minimum and maximum temperature grids of SPARTACUS and evaluated against station based FAO-Penman-Monteith estimates. The results indicate a considerable reduction of the bias particularly during winter across all levels of altitude and during summer, especially at higher elevated locations between 500 and 1000 m a.s.l. (c.f. Figure 12 in Haslinger and Bartsch, 2016). Averaged over all stations where Penman-Monteith AED is available (42 in total) monthly mean biases range between -0.17 mm day$^{-1}$ (February) and +0.80 mm day$^{-1}$ (April) and Root Mean Squared Errors are largest in June (1.42 mm day$^{-1}$). However, calculating the reference data using station time-series, only shortwave net radiation was considered. Omitting the mainly

outgoing longwave radiation leads to an overestimation of available energy on the surface and thus, an overestimation of potential evapotranspiration. To account for this incorrect representation of the energy balance in the initial ARET dataset, correction fields were applied. These were derived as the expected value (median per day of the year) of daily differences from 2013 to 2021 to Penman-Monteith reference evapotranspiration fields based on INCA input fields (Haiden et al. 2011), also considering outgoing longwave radiation.

A crucial part in this assessment is the observed trend of AED with respect to the given changes in atmospheric forcing over the reference period. In a recent study by Duethmann and Blöschl (2018) the authors estimated an annual Penman-Monteith AED trend across many river catchments in Austria of $18 \pm 5$ mm year$^{-1}$ decade$^{-1}$ for the period 1977-2014. Furthermore, they concluded that nearly 80% of the observed trend is attributable to changes in surface radiation, whilst temperature changes forced 20% of the trend. Changes in specific humidity and wind speed had no impact in observed AED trends. When using the ARET dataset for the entire Austrian domain the trend of annual AED sums from 1977-2014 is $17.8 \pm 3.0$ mm year$^{-1}$ decade$^{-1}$. We furthermore assessed the relationship between changes in AED and temperature, applied both for the observational and scenario data. The temperature trend over the entire Austrian domain from 1977-2014 is +0.47 °C decade$^{-1}$ (SPARTACUS data), which relates to an AED trend of 17.2 mm year$^{-1}$ decade$^{-1}$ (see above). This yields an AED increase of +36.6 mm year$^{-1}$ °C$^{-1}$. For the climate scenarios, based exemplarily on RCP4.5, from 2010-2050 a temperature increase of +0.28 °C decade$^{-1}$ is apparent, compared to an AED increase of +10.1 mm year$^{-1}$ decade$^{-1}$. These results indicate a scaling of +36.1 mm year$^{-1}$ °C$^{-1}$ of AED with a given temperature forcing, which is in very close agreement with the observed value of 36.6 mm year$^{-1}$ decade$^{-1}$. These results of the temperature scaling and the good agreement of the observed trends between AED of Duethmann and Blöschl (2018) and the one following the approach of Haslinger and Bartsch (2016) using a re-calibrated Hargreaves formulation proves that this simpler AED method is able to provide a physically sound representation of the main processes driving changes in AED.

### 3.3 Snow accumulation and snow melt

SNOWGRID (Olefs et al., 2013) is a physically-based and spatially distributed snow model, usually applied for operational forecast and driven by gridded meteorological output from the integrated nowcasting model INCA (Haiden et al., 2011). Recently, a climate version of the model, SNOWGRID-CL (SG-CL) has been developed and was applied to historical gridded meteorological data (SPARTACUS) in Austria (Olefs et al., 2020). SG-CL uses an adapted and extended degree-day scheme based on Pellicciotti et al. (2005) to calculate snow ablation, accounting for air temperature and the shortwave radiation balance. The latter is calculated from clear-sky solar radiation model output, a cloudiness correction based on the diurnal temperature range as well as surface albedo (weighted average of snow) and snow-free albedo using CORINE land cover types and related values given in the literature. The actual incoming shortwave radiation is computed as a product of clear-sky incoming shortwave radiation and a cloud transmission factor, representing the attenuation of solar radiation by clouds. The clear-sky incoming shortwave radiation is calculated as the sum of direct, diffuse and reflected shortwave radiation and requires knowledge of the exact position of the Sun and its interaction with the surface topography, as well as the transmissivity of the

atmosphere (Olefs et al., 2020). This snow ablation scheme is especially appropriate for climatological simulations (historical runs and future scenarios) as several studies showed their temporal robustness (Gabbi et al., 2014; Carenzo et al., 2009) which is key for a vigorous trend analysis. Snow accumulation is derived from daily fresh snow water equivalent taken as the solid fraction of daily precipitation sum. The solid fraction of precipitation is calculated using the daily average air temperature in a calibrated hyperbolic tangent function. Snow sublimation is calculated from daily potential evapotranspiration fields (Haslinger and Bartsch, 2016) using precipitation as a dampening factor. It uses a simple 2-layer scheme, considering settling, the heat and liquid water content of the snow cover and the energy added by rain (Olefs et al., 2013). Precipitation undercatch is corrected for and a simple scheme that accounts for the effect of lateral snow redistribution. Herein, SG-CL is driven by gridded observations and the historical simulations of OEKS15 for the reference period and with scenario simulations of OEKS15 considering near and far future time periods.

### 3.4 Glacier runoff

In order to assess the changing impact of glacier melt on water resources we apply the GLOGEM model results from Huss and Hock (2018, 2015) to all Austrian glaciers that are included in the Randolph Glacier Inventory V6.0 (RGI2017) for the scenarios RCP2.6 and RCP4.5 on a monthly basis. GLOGEM computes glacier mass balance and associated geometry changes for each glacier individually as described comprehensive in Huss and Hock (2015) and Huss and Hock (2018). The climatic mass balance is calculated at a monthly resolution based on near-surface air temperature and precipitation time series. Total mass changes are used to adjust each glacier's surface elevation and extent on a yearly basis using an empirical parameterization (Huss et al., 2010). We use their discharge product that accounts for changing glacier area and derive the rate of changing area from the model output of the same source for consistency. It explicitly represents the runoff that is made available from the melted ice volume (Huss and Hock 2015). We then accumulate time series of total discharge for all glaciers in Austria and derive specific discharge for the entire (glacier and ice-free) area >2500 m a.s.l. ($2.308*10^9$ m²). 2500 m a.s.l. is used as a threshold for areas potentially impacted by glaciers as this is approximately the elevation above which glaciers can occur in the study area (Fischer et al., 2015). Temporal averaging of this value allows for assessing changes of specific discharge in mm/month for the future time periods with respect to the reference period. A negative value of this change means a reduction of discharge in the latter period.

### 3.5 Methods of analysis

### 3.5.1 Climate Change Signal

In this paper we assess future changes by two metrics. First, the absolute change of a variable in the future compared to a reference period, which we refer to as the Climate Change (CC) signal. It is given by the difference between a future and a reference time period of a given variable. In this paper we define the period 1981-2010 as the reference period and distinguish

between a near future period: 2021-2050 and a far future period: 2071-2100. All CC signals are calculated as absolute differences on a monthly, seasonal or annual basis and either displayed spatially (maps) or aggregated to spatial means following the defined classes of elevation.

### 3.5.2 Frequency Analysis - Return Periods

As a second metric we use the concept of return periods to assess changing probabilities of drought occurrences under future climate change. As in classical extreme value statistics when the data are sampled as an annual series, the return period is defined as the inverse of the occurrence probability of an event. Traditional applications of frequency analysis in hydrology and meteorology considered upper extremes such as floods or heavy precipitation events, where the return period is defined as the inverse of the exceedance probability of the event. For the case of drought magnitude of the CWB we are interested in quantities at the lower tail of the distribution. We therefore estimate the return period of a given event as the inverse of its non-exceedance probability, in analogy to low flow events (e.g. Coles, 2001; Laaha et al., 2017).

The calculation of CWB return periods follows the general approach of statistical frequency analysis, where a theoretical distribution is fitted to the empirical distribution of the data to provide a robust estimate of the probability of events. As the CWB is a random variable which is unbound in the direction of both extremes, we assume a normal distribution to be a reasonable model. The model is fitted using the L-moments approach, which provides a robust estimate of model parameters in the case of outliers and observation uncertainty (Hosking, 1990).

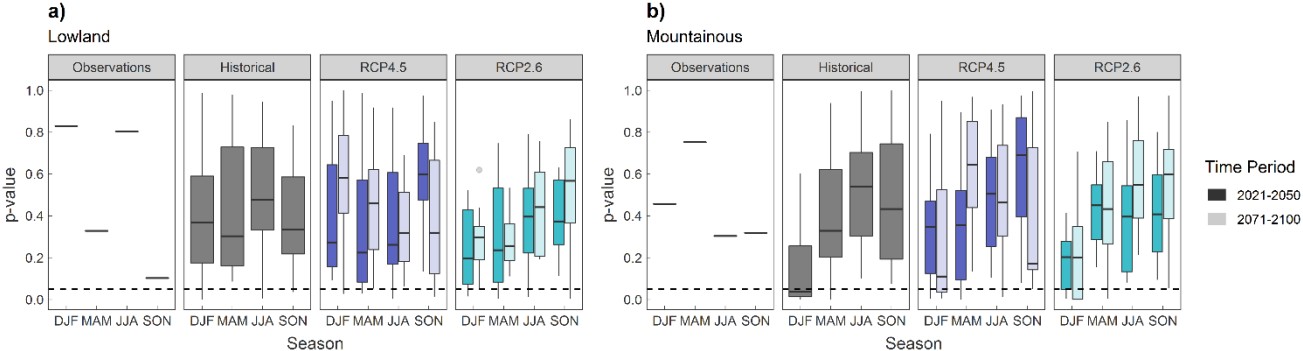

**Figure 2: p-values of the Shapiro-Wilk test for normality considering seasonal CWB values for a) lowland and b) mountainous areas for observations (based on SPARTACUS and ARET data) shown as short segments and the historical runs of the selected RCMs as well as for scenarios RCP4.5 and RCP2.6 for the near future (2021-2050) and the far future (2071-2100) displayed as boxplots denoting for the distribution among the different model realizations.**

We tested the assumption that the annual series of CWB indices (of different seasons, and stratified by lowland and mountainous areas) follow a normal distribution, using the Shapiro-Wilk test for normality (Shapiro and Wilk, 1965). The Null-hypothesis is that the data sample is following a normal distribution, p-values below the 5% threshold indicate that the Null-hypothesis is rejected and that the data is coming from another distribution.

The respective p-values are displayed in Figure 2 for the lowland (2a) and the mountainous (2b) areas for observations and RCM simulations. Considering the observations, the p-values are mostly well above the 5% levels, only autumn in the lowland shows a p-value closer to the significance threshold. Considering the climate simulations during the historical period there is a considerable spread of p-values among the different models, some of them even below the 5% level. Particularly during winter in the mountainous areas half of the model ensemble the distribution is most likely not normal. However, in general median p-values are ranging between 0.3 and 0.6 indicating normality for the majority of the model runs. During the scenario time periods, a similar picture emerges, median p-values are way above the 5% significance level. However, again in winter in the mountainous areas there is a considerable number of models with p-values below the 5% level. Although the CWB of some model runs is most likely not following a normal distribution as observed, the majority of the simulations does and therefore enables a direct comparison of distribution features between the reference and future time periods. For the winter, higher uncertainties have to be taken into account.

Similar results are obtained assessing the stationarity of the different 30-year time periods considered, which is a general assumption of classical frequency analysis (Coles, 2001). We tested the observations as well as the climate scenario time periods for significant trends as an indication for non-stationarity. In the observations all 30-year time periods investigated (for each season and for lowland and mountainous areas) do not show significant trends of the CWB following the Mann-Kendall trend test. These results are in line with similar investigations by Blöschl et al. (2018) who could show that increasing AED is balanced by an increase in precipitation. Considering the climate simulations, 13% of all individual 30-year periods (576 in total given by 24 individual runs times 4 seasons times 3 time periods times 2 different areas) show significant trends at the 5% level, thereby indicating non-stationarity. However, since this is only a minor fraction, and 30-year time-slices are relatively short for assessing the stationarity of climate simulations, we consider classical extreme value theory to be generally applicable, while the related uncertainties are taken into account when interpreting the results.

For assessing changing probabilities of extreme drought events under climate conditions we examine future changes on return periods for a given event threshold. At first we use a 10-year event return period under historical climate conditions as a reference. We fitted a normal distribution to the historical climate simulations using L-Moments for obtaining the distribution parameters. Then the same procedure is carried out for the future climate simulations, however this time we used the 10-year event threshold from reference period to estimate the return period for this event from the fitted distribution. This yields the change in return period of a 10-year event und future climate conditions. We applied the same method for assessing the change in event return period of the 2003 summer drought event. This severe event is still a benchmark in terms of severity considering the past centuries (Laaha et al., 2017; Haslinger et al., 2018a; Haslinger and Blöschl, 2017; Ionita et al., 2016).

# 4 Results

## 4.1 Future change in average climatic water balance conditions

Average annual and seasonal CWB values over the Austrian domain from observations and respective CC signals are summarized in Table 2. During the reference period 1981-2010 the annual CWB from observations is +466 mm year$^{-1}$, in
winter lowest values are apparent (+42 mm season$^{-1}$) due to lower precipitation rates in general and the build-up of snow pack. In the transition seasons spring and autumn values are rather similar with +110 and 133 mm season$^{-1}$ respectively. Largest values of the CWB is apparent during summer with an average of +181 mm season$^{-1}$.

For the future periods the CWB is expected to increase in winter, with a larger increase for RCP4.5 (+30 mm season$^{-1}$ in the near future and +50 mm season$^{-1}$ in the far future) compared to RCP2.6 (+25 mm season$^{-1}$ in the near future and +32 mm
season$^{-1}$ in the far future). An increase is projected for spring as well, ranging between +17 and +26 mm season$^{-1}$ in RCP4.5 for near and far future respectively and are equal with +21 mm season$^{-1}$ for RCP2.6 in both future time periods. For these two seasons the ensemble spread is ranging roughly between 10 and 17 mm season$^{-1}$. For summer the CC signal is rather small, -4 and 0 mm season$^{-1}$ for both periods respectively in RCP4.5 compared to -5 and -2 mm season$^{-1}$ in RCP2.6. In contradiction, the uncertainty of this CC signal is rather large given the wide range of the ensemble spread which is specifically large in
RCP4.5 reaching CC signals of ±40 mm season$^{-1}$ during the far future period. The ensemble spread is much smaller in RCP2.6, which might be also related to the smaller number of individual model runs, but still the ensemble spread is one half to a third of the RCP4.5 spread. Autumn is showing a moderate increase of the CWB with +13 and +31 mm season$^{-1}$ for RCP4.5 and the near and far future periods and +19 and +12 mm season$^{-1}$ for RCP2.6 respectively.

**Table 2: Climate change signal of the climatic water balance, average values over the Austrian domain and ensemble spread (1 standard deviation of the ensemble distribution) on a seasonal (winter: DJF, spring: MAM, summer: JJA, autumn: SON) and annual (ANN) basis**

| | | Observations | RCP4.5 CC signal and uncertainty | RCP2.6 CC signal and uncertainty |
|---|---|---|---|---|
| | | mm season$^{-1}$ | mm season$^{-1}$ | mm season$^{-1}$ |
| DJF | 1981-2010 | +42 | | |
| | 2021-2050 | | +30 (±11) | +25 (±8) |
| | 2071-2100 | | +50 (±11) | +32 (±12) |
| MAM | 1981-2010 | +110 | | |
| | 2021-2050 | | +17 (±15) | +21 (±17) |
| | 2071-2100 | | +26 (±11) | +21 (±13) |
| JJA | 1981-2010 | +181 | | |
| | 2021-2050 | | -4 (±24) | -5 (±14) |
| | 2071-2100 | | +0 (±40) | -2 (±19) |
| SON | 1981-2010 | +133 | | |
| | 2021-2050 | | +13 (±15) | +19 (±19) |
| | 2071-2100 | | +31 (±13) | +12 (±18) |
| | | mm year$^{-1}$ | mm year$^{-1}$ | mm year$^{-1}$ |
| ANN | 1981-2010 | +466 | | |
| | 2021-2050 | | +56 (±30) | +61 (±30) |
| | 2071-2100 | | +107 (±56) | +63 (±34) |

On an annual basis the simulations project a wetter future, CWB is about to increase by +56 mm year$^{-1}$ and +107 mm year$^{-1}$

for the near and future time periods respectively under RCP4.5 while under RCP2.6 these values are somewhat lower with +61 mm year$^{-1}$ and +63 mm year$^{-1}$.

A spatial assessment of the CC signal of the CWB as well as its components (rainfall, snow melt and atmospheric evaporative demand) for both emission scenarios and future time periods is given in Figure 3. The changes in the CWB (Figure 3a) are rather heterogeneous in space, creating a diverse pattern. Under RCP4.5 in the near future we see slightly increasing CWB

north of the main Alpine crest whereas in the southern parts of the domain there is a signal of decreasing CWB apparent. This signal shifts towards the end of the century towards increasing CWB mostly over the entire domain, with exceptions in the western, central alpine parts of Austria. RCP2.6 shows a somewhat different response, with increasing CWB throughout the domain in the near as well as in the far future. Exceptions here are as well some parts in the western most areas of Austria showing slightly decreasing CWB.

Figure 3b shows the spatial patterns of changes in rainfall (liquid precipitation), which is generally increasing across both scenarios and both future time periods. However, subtle differences are apparent, for example in RCP4.5 the increase in rainfall is larger by the end of the 21$^{st}$ century compared to the near future, and whilst the southern areas are showing smaller CC signals in the near future, this is no more the case for the far future period. On the other hand, in RCP2.6 the CC signal is not changing significantly over the 21$^{st}$ century.

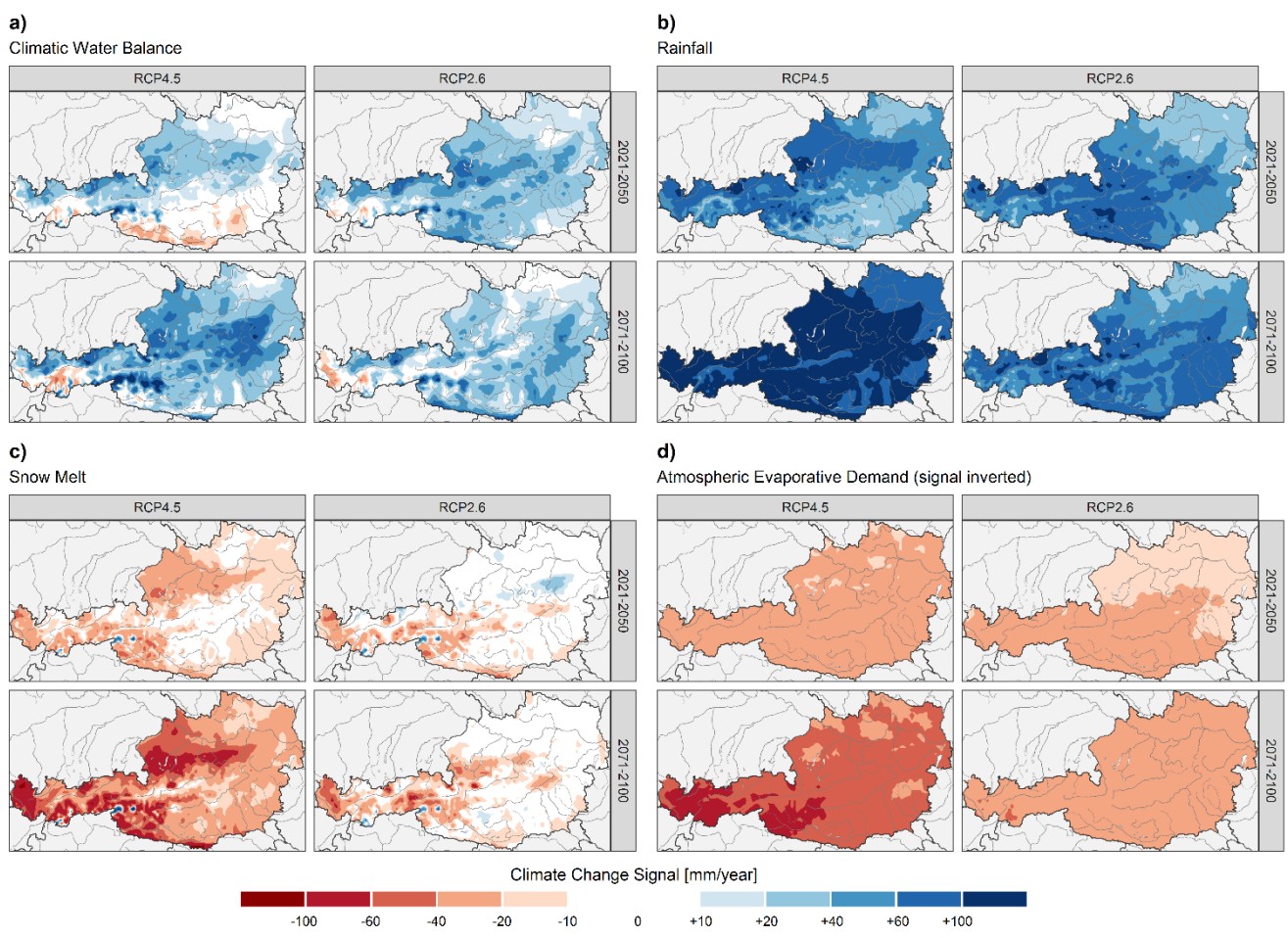

**Figure 3: Median ensemble climate change signal of RCP4.5 and RCP2.6 for the near future (2021-2050) and the far future (2071-2100) of (a) the mean annual CWB and (b-d) the mean annual components of the climatic water balance: (b) liquid precipitation, (c) snow melt and (d) AED (note that the signal is inverted, negative values indicate an increase in AED)**

For the changes in snow melt, rather different patterns emerge as displayed in Figure 3c. The overall temperature increase following future global warming leads to a subsequent reduction in snow melt. This is caused by a decreasing fraction of solid precipitation compared to the total precipitation sums and therefore a decreasing snowpack, which in turn is leading to declining snow melt. This CC signal is more pronounced in RCP4.5 following the stronger temperature increase. Considering spatial patterns, largest decreases are found in the Alpine fringes where precipitation in absolute terms is also highest (c.f. Figure 1c). Only in the high Alpine areas along the main alpine crest snowmelt is increasing, which is due to generally increasing precipitation and meteorological conditions still cold enough to build up a persistent snow pack during winter. In RCP2.6 these changes are similar in their spatial pattern, although smaller. There is nearly no CC signal in the lowland areas in the near and far future, exceptions are some subtle increases in some eastern mountainous areas, which are most likely due to increasing total precipitation in the region.

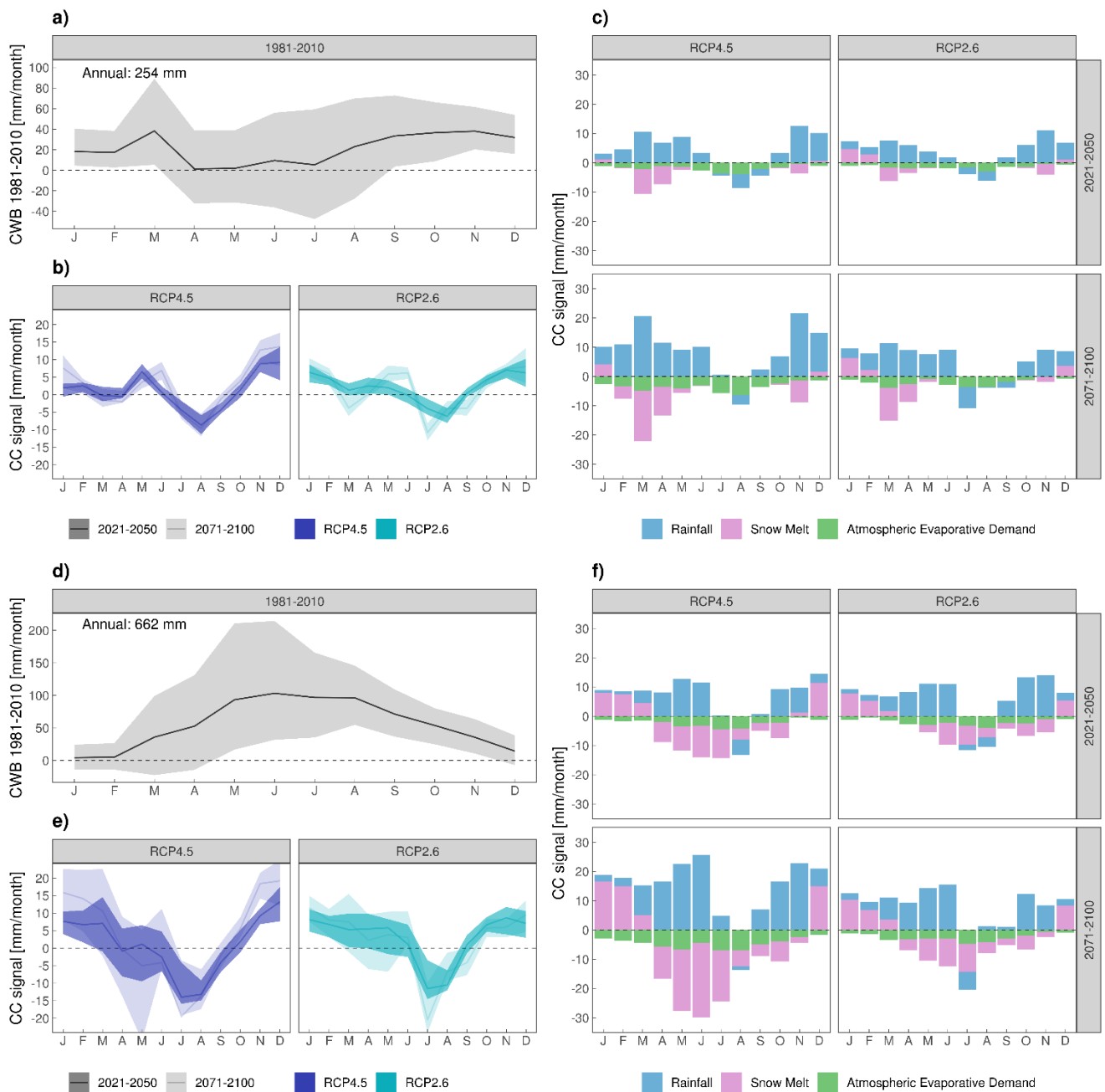

Figure 4: Monthly climate change signal of the CWB for lowlands in the upper panel and mountainous areas in the lower panel; (a,d) observed average monthly CWB in the reference period 1981-2010, where the shading denotes for the spatial variability of the CWB climatology, (b, e) ensemble median monthly climate change signal of the CWB for RCP4.5 (blue) and RCP2.6 (turquoise) for the near future in bold colour and the far future in pale colour, the shading denotes for the ensemble spread given by the 10th and 90th, (c, f) ensemble median monthly climate change signal of the individual CWB terms; rainfall: blue, snow melt: magenta, AED: green.

The CC signal of AED as displayed in Figure 3d and shows a more homogeneous pattern in space than the other variables. Smaller increases are visible in RCP2.6 due to the smaller temperature forcing. On the other hand, the signal is stronger in RCP4.5 with a slightly stronger signal in the mountainous areas.

In the light of this spatial assessment of changes in average CWB and its components it is important to consider seasonal variations of change as well. Figure 4 shows these with regards to lowlands and mountainous areas. The results for lowland areas are summarized in Figures 4a-c, where 4a displays the spatially averaged monthly mean CWB during the reference period 1981-2010 based on observations. It shows somewhat larger values during winter and autumn with a small snow melt induced peak in March, lower values are apparent from May to August. On average on an annual basis the CWB is +254 mm

year$^{-1}$ in the lowlands. Considering future CWB changes (Figure 4b) there is a mostly coherent CC signal of increasing CWB during the cold season months for both time periods and emission scenarios. An exception is early spring (March and April), where a negative CC signal is visible under RCP2.6 in the far future. Positive CC signals are apparent during the beginning of the warm season (May and June) as well, particularly under RCP4.5 (both time periods) and RCP2.6 (far future). On the contrary, negative CC signals appear during July, August and September (both time periods and emission scenarios) which are

largest mostly in August (-5 to -10 mm). The contributions to these changes from the individual terms of the CWB equation are displayed in Figure 4c. Two things are obvious at first sight, on one hand larger changes during the far future period and on the other hand slightly larger changes for RCP4.5, although foremost during the far future period. Biggest changes are apparent for rainfall, here a positive CC signal is seen during all months (largest during spring and autumn) except for July to September where negative CC signals are visible to some extent. Seasonally punctuated changes are visible for snow melt,

where positive changes are visible in the winter months (December, January, and February) and negative deviations mostly during spring. These are most likely caused by seasonally shifted snow accumulation and ablation processes with higher temperatures causing earlier snow melt which is lacking during those months were snow melt mostly occurred in the reference period. Reasons for increasing snow melt during winter might arise from higher temperatures as well, causing snow packs to more often melt during the winter months in future time periods than was observed in the reference periods. In contrast to

these rather large changes of these two variables, the CC signal of reference evapotranspiration is rather small. It is of course largest in the far future and RCP4.5 which shows a bigger temperature increase, however, the largest deviations are -5 mm month$^{-1}$ for July (RCP4.5, far future) which is considerably smaller than changes in rainfall and snow melt (up to +20 mm month$^{-1}$).

As is visible from Figure 4b the CC signal of the CWB is nearly zero for early spring (March and April), however, considering

the individual terms of the climate water balance huge dynamics are apparent with a considerable shift from snow melt to rainfall during this time of the year. Positive CWB signals are mainly caused by increasing rainfall, particularly during winter. On the contrary, the negative CWB changes during summer are caused by both increasing AED and slightly decreasing rainfall. Considering the mountainous part of Austria (Figure 4d-f) there is a completely different climatological initial condition. Figure 4d displays the average monthly CWB which depicts an inverted annual evolution compared to the lowland (c.f. Figure

4a) with lowest values during winter (slightly above zero) and largest values during summer (maximum in June, +100 mm). These low values during winter are mainly caused by strongly reduced positive moisture flux from rainfall and snow melt, since precipitation appears mostly in the form of snow which is accumulated and later in the year released through snow melt. These snow melt processes along with increasing precipitation sums in the warm season add up to the peak appearing in summer. On average the observed CWB on an annual basis is +662 mm year$^{-1}$.

The CC signal of the CWB is displayed in Figure 4e. For both scenarios and both future time periods a similar pattern is apparent, showing increasing CWB during the cold season, particularly during winter. Differences arise during spring, where RCP4.5 is showing no clear change whereas in RCP2.6 an increase is visible as large as during the winter months. Common in both scenarios is the distinct negative CC signal during July and August where negative deviations between -10 and -15 mm month$^{-1}$ occur. The patterns of change of the individual terms of the CWB are depicted in Figure 4f.

In general, these patterns are similar to the lowland, however the magnitude of the CC signal is much larger and there are also larger changes for RCP4.5 and the far future period. The mountainous areas show a pronounced increase in rainfall similar to the lowlands, with highest CC signals from May to June and October to November. As for the lowland areas slightly negative changes are visible in the summer months. In the higher elevated regions the impact of snow accumulation and ablation is far bigger compared to the lowlands, hence there are considerable changes in snow melt apparent over the course of the year. In

particular there is an increase in snow melt from December to March, again largest in RCP4.5 and for the far future. During the remaining months the future CC signal is negative, most pronounced during June and July. This points to a shift of the strongest seasonal CC signal between lowland and mountainous areas. Here the signal is stronger later in the year, along with a generally later melting season. As for the lowland areas the contribution of changes in AED is small compared to the rainfall and snow melt components and is within a range of -5 to -10 mm month$^{-1}$ during the summer months.


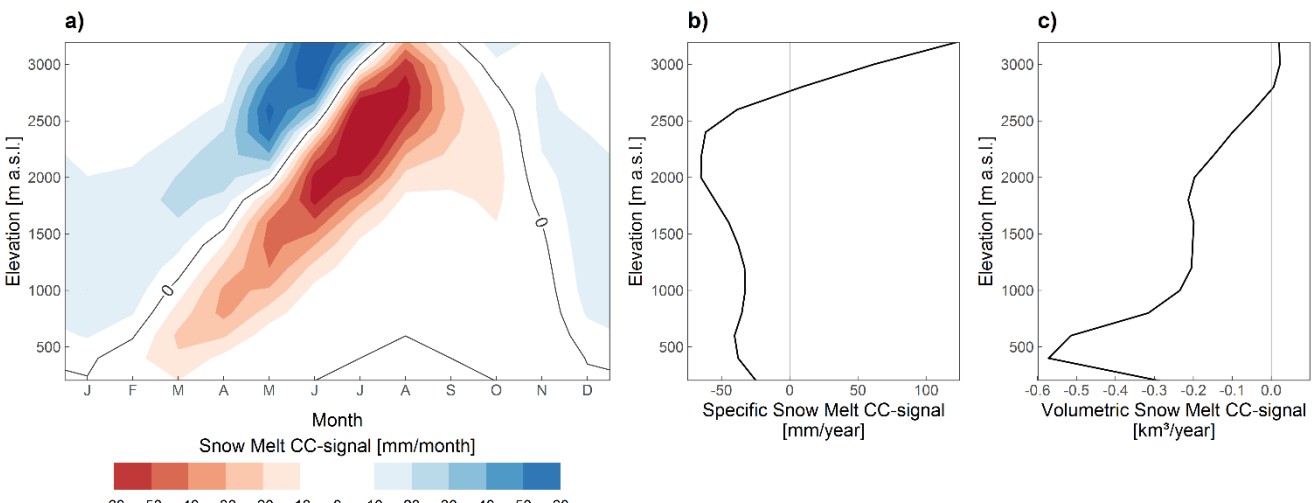

**Figure 5: (a) Snow melt CC signal depending on time of the year (month) and elevation for RCP4.5, 2071-2100, (b) averaged over all months (c) given as volumetric change (multiplied by area of elevation class).**

Given this detailed analysis of future changes of the individual components of the CWB in lowland and mountainous areas it is apparent that snow melt changes may exhibit the largest changes across seasons and elevation bands. To shed more light on this matter Figure 5a shows exemplarily the CC signal of snow melt for RCP4.5 and the far future for the individual months and from 200-3400 m.a.s.l. elevation. Here we see a general increase in snow melt during winter (DJF) between 500 and 2000 m a.s.l. of 10-20 mm month$^{-1}$. However, the CC signal in both the negative and positive direction is getting stronger during spring and summer. A distinct dividing line along season and elevation is apparent, separating elevations with negative and positive snow melt CC signal. The magnitude increases as well with elevation which is due to the increasing total precipitation sums at higher elevations. For every point in time of the spring/summer season there is a critical level of elevation with zero change and positive CC signal above that and vice versa. Increasing snow melt above the critical elevation is caused by both increasing precipitation during winter and spring and higher temperatures causing more snow melt than in the reference period. On the other hand, decreasing snow melt below the critical elevation is due to thinner snow pack following decreasing snow accumulation during winter and a higher rainfall fraction along increasing temperatures.

On average over the course of the year (Figure 5b) snow melt is decreasing up to 2700 m a.s.l., higher up, snow melt increases. This pattern is driven by increasing temperatures leading to less snow accumulation, particularly in lower elevations. However, the signal changes at higher elevations (>2700 m a.s.l.) where snow melt is increasing. This is due to the increasing total precipitation amount during winter, and still low enough temperatures to build up a significant snow pack which is the reason for the positive snow melt signal. Assessing these changes in a volumetric perspective (multiplying by the areal extent of the elevation band) gives a rather different picture (Figure 5c), where largest changes are found below roughly 700 m a.s.l., due to the larger spatial extent, highlighting that these areas are most sensitive to snow melt changes in absolute terms.

A special case in this assessment of CWB changes across Austria is the spatial domain of the high alpine areas (> 2500 m a.s.l.) due to the considerable fraction of these covered with glaciers. The seasonal evolution of the CWB in the high alpine domain is displayed in Figure 6a. During the cold season from November to April/May the CWB is slightly negative. Most precipitation occurs in the form of snow, consequently building up the snow pack which acts as a storage term for the summer months. The slightly negative values during winter are due to the small but steady losses due to AED. However, from May onwards the snow melt season sets in and also the fraction of liquid precipitation increases, leading to a steep rise of the CWB until its peak in July (+200 mm month$^{-1}$) before it approaches zero again in October. The average CWB is 413 mm year$^{-1}$.

Future changes of the high alpine CWB are displayed in Figure 6b. The patterns are similar in both scenarios showing hardly any change until May, were positive CC signals are visible. The CC signals are getting strongly negative during July, August and September and are only minor for the rest of the year. A major difference compared to the lowland and mountainous change patterns (c.f. Figure 4b and 4e) is the stronger CC signal during the far future period. In the high alpine area the CC signal is more pronounced than in the lowlands and mountainous areas. The reduction of the CWB is around -100 mm month$^{-1}$ for July and August for the far future period, which is a reduction of 50% compared to the CWB in the reference period 1981-2010 (c.f. Figure 6a).

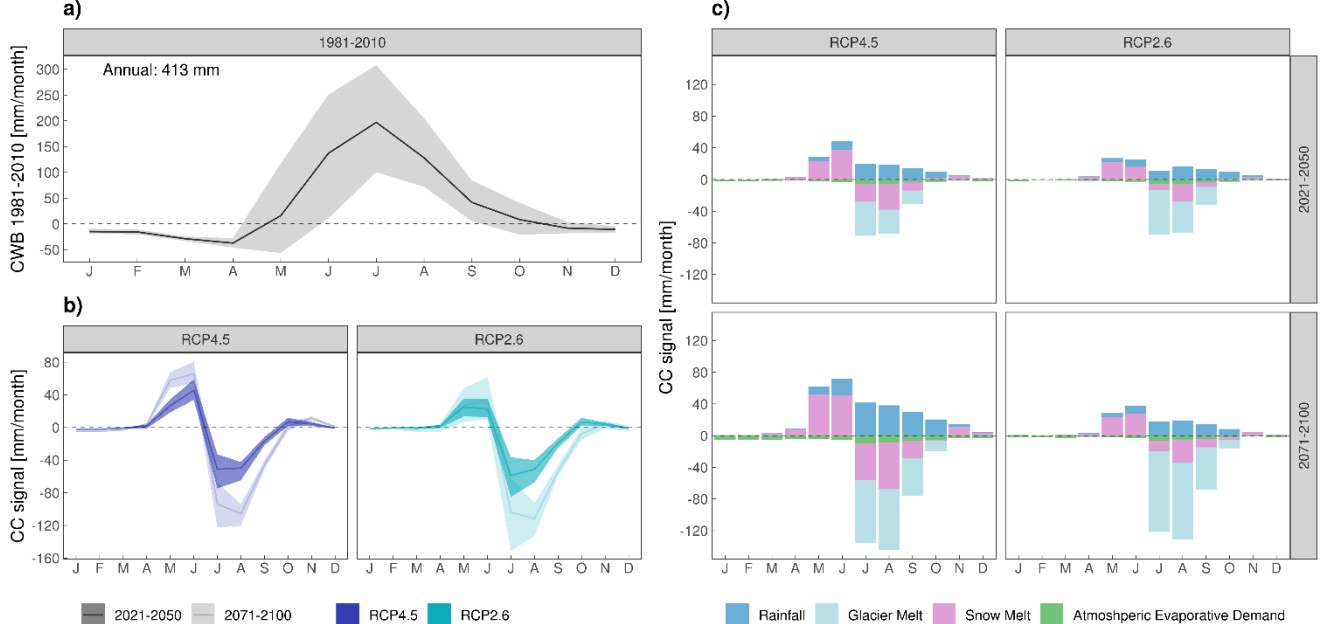

**Figure 6: Monthly climate change signal of the CWB for the high alpine areas; (a) observed average monthly CWB in the reference period 1981-2010, where the shading denotes for the spatial variability of the CWB climatology, (b) ensemble median monthly climate change signal of the CWB for RCP4.5 (blue) and RCP2.6 (turquoise) for the near future in bold colour and the far future in pale colour, the shading denotes for the ensemble spread given by the 10th and 90th, (c) ensemble median monthly climate change signal of the individual CWB terms; rainfall: blue, snow melt: magenta, AED: green.**

The reason for these large changes is revealed by examining the change of the individual components of the CWB (Figure 6c). In addition to the three main components of the CWB (rainfall, snow melt and AED) we consider glacier melt for the high alpine areas as well (see section 3.4 Glacier runoff for details). From May to October an increase in rainfall is contributing positively to the CWB CC signal, which is strongest in RCP4.5 in the far future with +40 mm month$^{-1}$ in July and August. In addition, snow melt increases during May. On the other hand, snow melt is considerably decreasing from June to September, again most pronounced in RCP4.5 in the far future period. This pattern resembles that of the mountainous areas, although the peak of the negative CC signal is in August for the high alpine area. Similar processes cause these changes, namely reduced snow pack during summer due to earlier ablation under a warmer future climate. The most important driver of the negative CC signal of the CWB is the change in glacier melt. It is the largest contributor from July to September and shows largest signals in the far future. Continued warming leads to sustained ice loss, which produces increasing runoff after initial temperature increase. However, once a critical threshold (commonly referred to as 'peak water') is exceeded, the runoff decreases due to the shrinking ice volume of the glaciers. By the near future period this threshold is most likely surpassed by all glacier covered areas in Austria; thus, decreasing glacier runoff is a consequence of further future warming (e.g. Huss et al. 2018, Pepin et al. 2021).

## 4.2 Future change in extreme drought event probabilities

Apart from assessing changes in the mean state of the CWB and its components on an annual and seasonal basis it is important to quantify changing probabilities of drought events of a certain threshold. As described in more details in the Methods section we define a moderate drought event where the CWB deceeds the 10-year return period threshold during the reference period 1981-2010. Now we estimated the return period of this given reference threshold for future climate conditions.

The results on a seasonal basis and stratified by lowland and mountainous areas are displayed in Figure 7. For winter an increase in return period (lower probability for drought occurrence) is given across all scenarios, time periods and elevation areas. However, the signal is stronger in RCP4.5 with a median return period across all models of around 20 years (lowland and mountainous) for the near future and 30 years for the lowland (50 years for mountainous) for the far future. The signal in RCP2.6 is less pronounced with future return periods between 12 years (near future) and 20 years (far future), similar across lowland and mountainous areas.

Only subtle changes are apparent in spring. Here the median future return periods are only marginally lower, here particularly under RCP2.6 in the lowlands. During summer, median return periods for the given threshold are generally decreasing in both future time periods, scenarios and elevation regions. However, the signal is more robust in the near future, due to smaller ensemble spread, here a median return period between 7 years (RCP4.5) and 9 years (RCP2.6) for the lowland and 6 years (RCP4.5) and 7 years (RCP2.6) for the mountainous areas are apparent. For the far future, uncertainty is increasing depicted by increasing ensemble spread, median return periods range between 4 and 9 years.

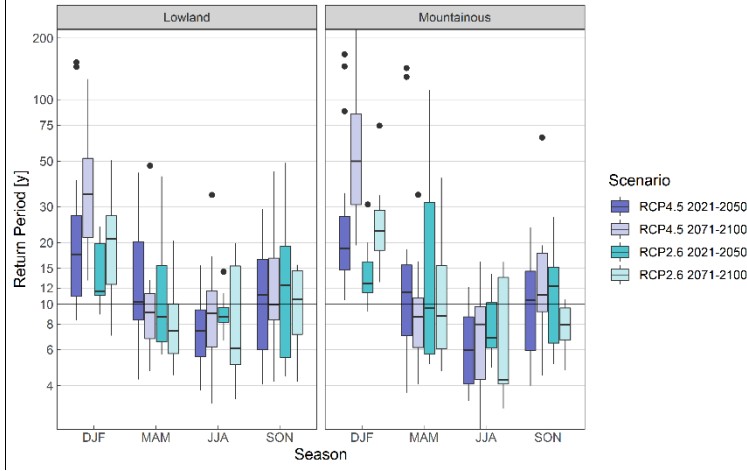

**Figure 7: Change in return period of a CWB 10-year event in the reference period (1981-2010) for lowlands (left) and mountainous areas (right) under RCP4.5 (blue) and RCP 2.6 (turquoise) for the near future (2021-2050, bold colours) and the far future (2071-2100, pale colours), the boxplot denote for the ensemble spread of the individual climate model runs.**

In autumn no clear change signal is visible, median return periods are marginally increasing in both time periods, scenarios and elevation areas (one exception is RCP2.6 in the far future in mountainous areas), however uncertainties are again large due to substantial ensemble spread.

The projected changes point towards subtle decoupled signals of the mean change and changes in the tails of the distribution. While there is no clear sign for changes in the mean CWB values in summer (c.f. Table 2) we see here some evidence for increasing probabilities for moderately extreme drought conditions during summer. To examine the future risk of extreme

drought conditions in more detail, we analyse the drought event of the year 2003 in the light of its return period for past climate conditions and future projections. This event was extraordinary in terms of its magnitude, spatial extent and impact (Black et al., 2004; Schär and Jendritzky, 2004; Fischer et al., 2007) and apart from other events that struck Austria in recent years (drought of 2015, e.g. Laaha et al., 2017; Ionita et al., 2016 and 2018, e.g. Buras et al., 2020), the event of 2003 impacted Austria nearly entirely. Other reported events affected only the northern parts of the country highlighting the prominent dipole

structure of drought events in the Alpine region (Haslinger and Blöschl, 2017; Haslinger et al., 2018b). In a first step we assess the severity of the event by estimating the return period of the CWB for summer (JJA) as well as for spring and summer (spring/summer, MAMJJA). Several publications highlighted the special precondition of the extraordinary dry spring which most likely intensified the consequent summer drought and heat wave (Haslinger et al., 2018a; Laaha et al., 2017; Fischer et al., 2007), this is why we chose to analyse summer season and the spring/summer season separately.

Figure 8a displays the grid-point-wise estimation of the 2003 event return period of the CWB for summer and spring/summer for the reference period 1981-2010. The maps indicate a higher severity if spring/summer is considered, which is also in line with other findings which claimed that the initialization of the drought was already in March (Haslinger and Blöschl, 2017; Fischer et al., 2007) starting into the warm season with a considerable moisture deficit. Also common in both maps is the higher severity in the western and northern parts of the country, the southernmost part was not that severely struck, particularly

if only summer is considered.

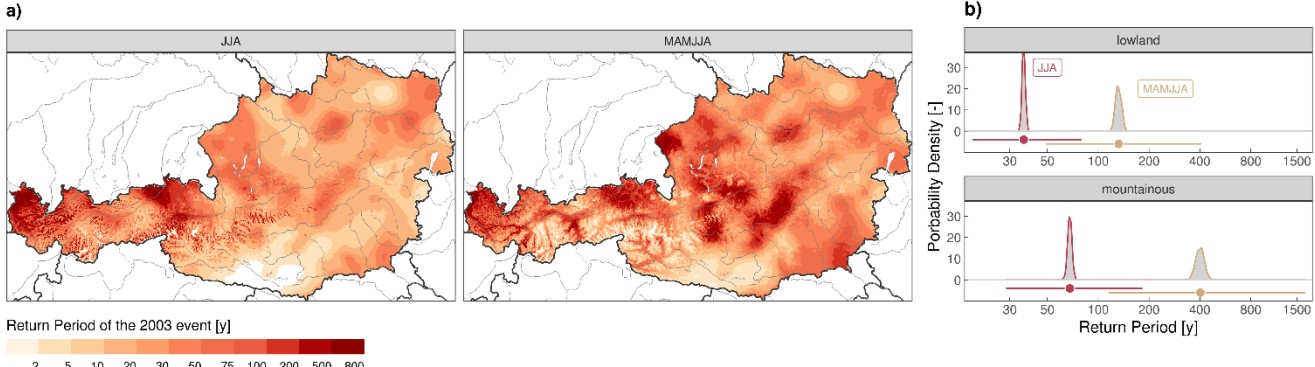

**Figure 8: (a) Spatial estimate of the 2003 event return period for summer (JJA) and spring and summer (MAMJJA) based on the observed CWB distribution in the reference period 1981-2010, (b) uncertainty assessment of the return period estimates; density**
**plots denote for the uncertainty from spatially averaging; a bootstrapping approach with 1000 iterations randomly sampling 10% of the respective grid points in the region and averaging afterwards is applied; point-range plots denote for the uncertainty from distribution fitting of the randomly sampled spatial averages, the range denotes for the full range of iterations, the dot for the median of the iterations for return period estimates.**

When deriving spatial averages for the lowland and mountainous areas of these return periods we assess the uncertainty arising
from spatial averaging and from distribution fitting. The results are shown in Figure 8b. The two panels show the uncertainty

assessment for the lowland and the mountainous areas, where the kernel densities are displaying the uncertainty from spatial averaging and the point range for the distribution fitting. In general, the uncertainty from the fitting is much larger compared to the spatially averaging, which is true for both areas. For the lowland areas the spatial uncertainty is between 34 and 38 years (117 and 152 years) for summer (spring/summer) and for the mountainous areas it ranges between 62 and 73 years (322 and

500 years) for summer (spring/summer). The uncertainty from distribution fitting is considerably larger, here the estimates range for the lowland between 18 and 90 years (50 and 400 years) for summer (spring/summer) and for the mountainous areas between 29 and 183 years (115 and 1673 years) for summer (spring/summer). The best estimate for the lowland is 36 years (132 years) for summer (spring/summer) and for the mountainous areas 68 years (403 years) for summer (spring/summer).

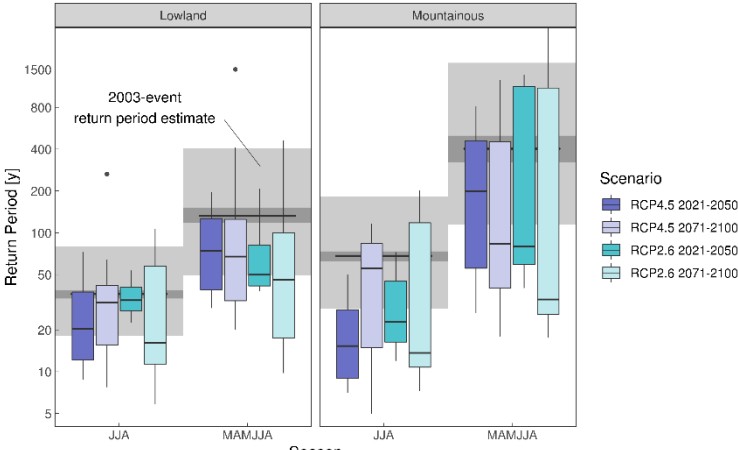

**Figure 9: Future return period estimate of a drought event with a 2003 event threshold for lowlands (left) and mountainous areas (right) under RCP4.5 (blue) and RCP 2.6 (turquoise) for the near future (2021-2050, bold colours) and the far future (2071-2100, pale colours), the boxplot denote for the ensemble spread of the individual climate model runs), the grey areas indicates the uncertainty from observed return period estimates (c.f. Figure 8b) in the reference period 1981-2010 where the darker shades indicate the spatial averaging uncertainty and the lighter shading the distribution fitting uncertainty.**

Similar to the previous analysis of future return periods of a 10 year event in the reference period we assess the return period for a 2003 event under future climate conditions (Figure 9). For the lowland in summer there is a reduction in return period visible, under RCP4.5 the ensemble median is around 20 years for the near future and higher in the far future, compared to 36 years in the reference period. Under RCP2.6 a reduction in return period (ensemble median) to 33 years (16 years) in the near future (far future). The decrease in return period is visible in the mountainous areas in summer as well with a drop to 16 years

(56 years) in the near future (far future) under RCP4.5. Under RCP2.6 the reduction is similar, here values of roughly 23 years (14 years) are apparent for the near future (far future). For lowlands in the combined spring/summer season (MAMJJA) changes are larger to some extent. The ensemble median is around 70 years for the near future and far future respectively under RCP4.5 and around 50 years for the near and far future respectively under RCP2.6. For the mountainous areas in spring/summer the ensemble median return period is ranging between 200 (RCP4.5, near future) and 33 years (RCP2.6, far

future) In general, uncertainties expressed via the ensemble spread are increasing with the time horizon (larger for far future estimates) and are larger for the combined spring/summer season and the mountainous areas. This might be related to the

complex nature or drought generating processes in snow dominated areas, here particularly during spring and the mountainous part of the country.

The probability for a rather extreme event like 2003 is strongly increasing under future climate conditions, which is the case
for all time periods and emission scenarios indicated by a median return period in future time periods well below the best estimate of the 2003 event return period in the reference period. It also seems that the model agreement is stronger at this even more extreme region of the CWB distribution, since the return period of 2003 best estimate is often outside the interquartile range ("box" of the boxplots). This feature is not as clear for the 10 year event return period change (c.f. Figure 7) for summer. In the light of the minor seasonal CC signals and an increase in probability for reaching extreme dry conditions, these results
points towards a general increase in (interannual) variability of the CWB, particularly during summer (and spring) as this season is examined in more detail here.

However, to quantify potential changes in the interannual variability of the CWB the change in standard deviation of summer and spring/summer CWB values for the lowland and mountainous areas are calculated for future climate conditions compared to the reference period. The results are displayed in Figure 10. As expected, the ensemble median change of the interannual
variability across both seasons, scenarios and time periods is increasing. One exception is RCP2.6 for the near future summer season, here a slight decrease is visible. Although the uncertainties are large, given the wide range of the individual boxplots, the climate scenarios consistently point towards a more variable future on an interannual time scale and therefore an increasing probability for experiencing extreme hydrometeorological states. This alterations of the climate systems and the implications for drought hazard risk is shown in other studies, e.g. Ukkola et al., (2020) who showed that increasing interannual variability
of precipitation is the main driver of drought risk in Central Europe. On the other hand, declining mean precipitation is driving increasing drought probabilities in the Mediterranean region.

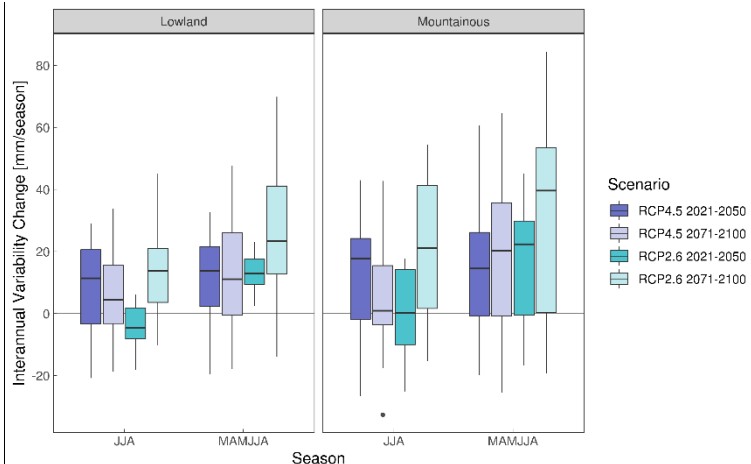

**Figure 10: Future change in interannual variability of seasonal CWB for lowlands (left) and mountainous areas (right) under RCP4.5 (blue) and RCP2.6 (turquoise) for the near future (2021-2050, bold colours) and the far future (2071-2100, pale colours), the boxplot**
**denote for the ensemble spread of the individual climate model runs.**

## 5 Discussion

In this study future surface water availability in the complex terrain of the Austrian domain is investigated as a function of elevation and climatic water balance components. Using two emission scenarios we found similar spatial pattern of the CC signals across the various variables investigated, although these are slightly stronger in the moderate mitigation scenario RCP4.5 due to larger temperature forcing. Here we use downscaled and bias corrected climate scenarios (RCP2.6 and RCP4.5) of the EURO-CORDEX initiative based on a tailored CMIP5 ensemble of Global Climate Models. We intentionally do not use an extreme emission scenario like RCP8.5, as the literature is growing on the implausible exaggeration of future emissions of this socio-economic pathway (see details in the data section). Although RCP4.5 might be seen as the most plausible scenario at current times, even larger global temperature increases cannot be completely discarded. Non-linear behavior and the potential of crossing tipping points of the climate system may cause additional warming, leading to temperature levels far beyond current plausible scenarios like RCP4.5. Although the existence of some tipping points and further the timing of them in general is still under debate (Brovkin et al., 2021; Lenton et al., 2008), uncertainties in the climate system arising from this matter have to be kept in mind.

Compared to former studies considering future drought conditions in Austria, slightly different results are obtained. For example, Haslinger et al. (2016) and Laaha et al. (2016) assessed future SPEI trends based on RCMs driven by AR4 CMIP3 GCMs. Here the authors found a general drying trend on an annual basis and for all seasons except for winter. This is contrast to our findings here, where the newer generation of models indicate wetter conditions in general on an annual basis and particular wetter winters and springs. This increase in precipitation is not only a feature in this study using this specific model ensemble, but also in several others (Kotlarski et al., 2023). The origins of this trend towards more wetness in winter and spring is not fully understood. However, Rajczak and Schär (2017) point towards the specific location of the alpine region in a transition zone between future precipitation increase (decrease) on the northern (southern) side of the main alpine crest.

In addition, we displayed future changes of the most important variables acting on the surface water availability. Apart from minor increases of AED which is also shown in Gali Reniu (2017) for the same scenario dataset, largest changes and shifts in seasonality are found for snow melt and the fraction of liquid precipitation. Our findings for Austria are in line with others considering different domains in the mid-latitudes (e.g. Musselman et al., 2021; Livneh and Badger, 2020, for the western United States) or on a global perspective with regards to agricultural drought (Qin et al., 2020). For the Central European mountain ranges the impact of snow melt changes on summer low flows was investigated by numerous studies (Meriö et al., 2019; Jenicek et al., 2018; Jenicek and Ledvinka, 2020) showing similar seasonal shifts. However, although some modelling uncertainties prevail (Olefs et al., 2020) we are confident that the presented results indicate robust snow melt CC signals given the projected precipitation and temperature changes.

In high elevated areas the contribution of glacier runoff to surface water availability becomes increasingly important (Kaser et al., 2010). For Austria we could show that changes in glacier runoff denotes for the largest fraction compared to the other climatic water balance components in the high Alpine areas during the summer months. The largest negative changes are found

in July and August, with positive changes, although smaller, occurring during May and June. This is in broad agreement with previous studies (e.g. Huss, 2011) which indicate that the so called "peak water" is already surpassed or will be in the near future, which means that the shrinking glacier volume along with warming is yielding less amounts of melt water. Although the contribution of glacier melt to the overall water balance of large river basins in Central Europe (e.g. Danube) is only minor during cool and wet summer season, it is strongly increasing during drought years, as particularly shown for the 2003 event (Huss, 2011). In the light of the results presented here, with "peak water" in the wake and increasing drought hazard risk in summer in future time periods is likely to stress low land water availability beyond experienced scarcities of the past.

Considering the occurrence of moderate (e.g. 10 year return period) and extreme (e.g. 2003 event) drought events in the future, the climate scenarios project an increase in return period (viewer events) during winter and no clear change in spring and autumn. However, during summer, a decrease in return period (more events) of moderate and extreme droughts is projected across all scenarios, although the mean CC signal of the CWB is around zero. The change is rooted in the increase of interannual variability of future climate, which is observable during all seasons. Increasing CWB during winter, spring and autumn however compensates the signal of increasing variability. Yet, this is not the case for the summer season, where rising variability significantly lowers the return period of moderate and extreme droughts. These findings are confirmed by another recent study investigating drivers of future meteorological drought changes for Central Europe (Ukkola et al., 2020). As thoroughly presented in Pendergrass et al. (2017) precipitation variability is expected to increase in a warmer climate across all time scales from days to decades. The authors pointed towards a nearly linear relationship between a temperature increase and the respective increase in atmospheric moisture content, precipitation variability and mean precipitation. Interestingly, variability change is in any case higher than a change in the mean, except for the winter season in the mid-latitudes and thus a rather robust signal of future climate change.

Although the results presented here show a more or less linear response of surface water availability and drought occurrence with regards to warming levels, it is important to interpret these findings in the light of observed drought variability. Numerous studies point towards a significant contribution of multi-decadal climate variability in driving drought periods in the Alpine Region and Central Europe (Hanel et al., 2018; Moravec et al., 2019; Haslinger et al., 2019a, b, 2018a). Internal variability of the climate system, predominantly in the North Atlantic ocean, is found to particularly drive changes in atmospheric circulation and thus drought favoring weather regimes over Central Europe (Haslinger et al., 2021; Sutton and Hodson, 2005; O'Reilly et al., 2017). However, current state-of-the-art GCMs still fail to reproduce the main features of multidecadal climate variability in the Northern Hemisphere (O'Reilly et al., 2021; Kravtsov et al., 2018). This might be due to lacking spatial resolution of the ocean and atmospheric models (Caesar et al., 2018), a problem which could be overcome in the near future with increasing computational power. It is important to understand the presented results in the light of climate scenarios which are driven by a merely linear anthropogenic greenhouse gas forcing of different magnitude. The response of the climate system over the course of the 21$^{st}$ century is tied to this forcing and, as mentioned, lacks to provide consistent multidecadal internal variability. To this end, it is very likely that multidecadal climate variability generates periods of, on one hand excess, but on the other

hand even lower water availabilities and hence drought conditions and in addition further climate change shifting seasonal regimes with the potential to drive water scarcity to unexpected levels.

## 6 Conclusion

In this study we presented scenarios of future surface water availability across different elevations and for the most important variables of the CWB. In general, wetter conditions are projected on an annual basis (e.g. RCP4.5 +107 mm, RCP2.6 +63 mm for the period 2071-2100), however particular seasonal shifts of snow melt and glacier melt at higher elevations change the surface water availability in the course of the year. Drought conditions in particular are expected to become more frequent during summer months, which is predominantly driven by an increase in interannual variability of the CWB. For a 2003-event

like situation during summer in the lowlands, the return period is estimated to decrease from 36 years to 20 years under RCP4.5 in the near future for example. From the knowledge of past drought periods, it is important to highlight the role of internal climate variability, which is not well depicted by current climate models. The climate projections are to be seen as potential conditions under a quasi-linear climate forcing from greenhouse gases. For the adaptation to future drought conditions it is therefore of utterly importance to prepare for more extreme states of the climate system which are not depicted by current

models. Local to regional efforts for storing water during times of excess and the re-cultivation of wetlands for example may be feasible measures in the short term having the potential to mitigate severe drought impacts. However, a deeper understanding considering the driver on a local to hemispheric scale of future water availability is necessary to apply sufficient adaptation to potential drought conditions and water scarcity.

## Code Availability

The complete code in R programming language is available upon request from the corresponding author.

## Data Availability

All observational data sets (SPARTACUS and ARET) are free to use for research purposes and available on the datahub of the ZAMG: https://data.hub.zamg.ac.at/; the Austrian reference scenario dataset OEKS15 is available at the data center of the Climate Change Centre Austria including the SNOWGRID-CL data: https://data.ccca.ac.at/

**Author contribution**

Conceptualization: KH; Data curation: KH, KA, MO, RK, JA; Formal analysis: KH; Investigation: KH, GL, JA, WS; Methodology: KH, KA, GL, WS, JA; Software: KH, KA, MO, RK, JA; Visualization: KH; Writing – original draft preparation: KH; Writing – review and editing: KH, WS, GL, KA, MO, RK, JA

**Competing interest**

The authors declare that they have no conflict of interest.

**Acknowledgements**

Some parts of the paper were supported by the European Union's Horizon 2020 programme under grant agreement No 776479 for the project CO-designing the Assessment of Climate CHange costs. https://coacch.eu/ and by the European Union's Alpine Space Program 14|20 under grant agreement No 940 for the project Alpine Drought Observatory.

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
