# Peer review of "Apparent contradiction in the projected climatic water balance for Austria: wetter condition on average versus higher probability of meteorological droughts."

_EGUsphere, 2022_

## Author Comment (AC1)

Dear Reviewer,

Thank you for your comments and suggestions. Our response is indicated below your comments in blue font.

**General comments**

This is valuable research, where changes in the climatic water budget (CWB) in the Alps, and future water scarcity risks are analyzed at an unprecedented spatial detail (1 km with a very good representation of mountain orographic factors).

It is also a work of great topical relevance and very timely, given the current 2022 heat wave in Europe and the water scarcity in the Alps.

This work explains well current and future alpine drought development mechanisms and shows how such events will become more likely in the future, despite possible year-round greater water availability in the Alps. Positive CWB signals are mainly caused by increasing rainfall, particularly during winter. On the contrary, the negative CWB changes during summer are caused by both increasing AED and slightly decreasing rainfall. Because of less snowpack contribution, water provision in the summer will become less constant and more connected to the liquid precipitation variability, and therefore, given the higher temperatures, summer water scarcity is more likely.

The figures are excellent ad full of information. Results are very interesting, especially for the spatial detail, which allows us to identify the local impacts of Alpine orography on the water budget. One relevant result is the finding that in the far future period the CC signal is much stronger in mountains than in lowlands.

I am looking for further research, showing how the 2022 summer drought will be compared to 2003 in terms of return period!

However, several methodological details need further clarification, especially on evaporation and glacier runoff estimation, and should be addressed before publication, as indicated below and in the specific comments.

- I am surprised by the strong increase of precipitation of the RCP 4.5 scenario for 2070 – 2100. Since most of the results are depending on this, and we know that climate models are still more uncertain in predicting precipitation trends than temperature, a discussion on the reliability of this high-precipitation RCP scenario would be very helpful.

  Yes, you are right, and thank you for your important comment! Precipitation projections are indeed highly uncertain, particularly in the Alpine region due to the complex generation of precipitation and the high fraction of convective precipitation. However, it is a feature seen also in other publications, e.g., https://doi.org/10.1002/2017JD027176. Here, an increase in precipitation in winter (+17 mm/d), spring and autumn (+4 mm/d) is contrasted by a decrease in summer (-4 mm/d), for RCP4.5 in the far future (2070-2099), which in total points towards more precipitation on an annual basis. This is in line with our study.

We will definitely discuss this feature in more detail and add the relevant literature (see above).

- I am also a little surprised you do not find an important increase in Evaporative Demand. I am just wondering if this could be related to the quite simplified approach used to estimate evapotranspiration ET. This point deserves further discussion.

  Yes, we will further discuss this crucial aspect of the paper in more detail. See particular suggestions below.

- **Title and journal.** The title triggers curiosity, but then reading the paper I found much more than this. The best parts of the paper are the high-resolution scenarios of climatic water balance (CWB) for Austria. So, the title is a little bit limiting. Changes in drought risk are only one of the aspects. I am also wondering if this paper fits better with HESS instead of NHESS.

  Thank you for your kind comment on the paper. You are right, there is more in it than just the drought aspect. However, we think that all the other topics rotate around the drought topic, which is from our perspective the most important point of the paper. This is the reason we choose to publish in NHESS, whereas HESS would also make perfectly sense. We do not know if a change in journal would be easily to accomplish and/or require an entirely new submission. In the interest of time, we thus for now prefer to stick to NHESS.

**Specific comments**

**Abstract.** Please specify that you consider only meteorological drought.

Yes, good point. We will specify that we are explicitly considering meteorological drought.

1. **Introduction (and title).**

L68 "How will future surface water availability change" I am not sure if the term surface water availability is correct. You are working with RCMS and a very simplified hydrological representation of the processes. What you get is mostly the water budget. Then you work on the CWB, which is simply a good indicator of the real surface water availability. Why do not also refer to CWB in the title?

Ok, thank you for your comment. It is true, that we do not fully represent the surface water availability, so referring to the CWB in the title would be more appropriate.

To have an estimation of surface water you need to model also water infiltration, vadose zone hydrology, and groundwater recharge losses… which is of course beyond the purpose of the paper.

**2 Data**:

L80 "The broadly used RCP8.5 scenario is intentionally not included here, since its emission pathway is highly unlikely from today´s emissions trajectories,"

Sure? Are you so optimistic about the future?

The literature on criticism considering overemphasising the RCP8.5 scenario is constantly growing. To make the results not too overloaded we decided to display RCP4.5 and RCP2.6 as one pathway which is optimistic and plausible and another one that is more optimistic. We will therefore better argue on limiting to those two, also highlighting the latest literature on RCP scenarios. Furthermore, we will add to the discussion an assessment on potential RCP8.5 implications to broaden the spectrum of the paper.

L87 "EURO-CORDEX downscaling," Could you provide more information on the approach? How is orography taken into account for the downscaling?

Thanks for your comment, we will discuss the downscaling procedure in more detail. Orography is indirectly taken into account, since the reference data (SPARTACUS) does explicitly consider orographic effects (cold air pools, inversions etc.) see these two publications for details:

https://doi.org/10.1007/s00704-017-2093-x

https://doi.org/10.1007/s00704-015-1411-4

**3 Methods**:

L119 "snowmelt model," Ok it is essential to consider snow melt in the CWB, but then you need a good snow melt model. Please provide here or later more information on how snow melt is modeled.

Yes, good point, we will add this rather relevant information in the methods section. We will add additional information of the extended degree-day approach described in Olefs et al. (2020).

"**3.2 Atmospheric Evaporative Demand**" AED is a key parameter. For this is essential the calculation of ETP. Which is the accuracy of the calculated ETP? How has been validated?

Yes, AED calculated by the given formulation of a re-calibrated Hargreaves equation and is evaluated in the referred publication:

https://doi.org/10.5194/hess-20-1211-2016

In comparison with station data Penman-Monteith (nearest neighbour gridpoint) and a Penman-Monteith estimate using data from the Austrian nowcasting system INCA the given method (ARET) is performing rather well. See this table from the publication indicated above:

**Table 2.** Error characteristics of ARET and INCA against station data.

| | Bias [mm day$^{-1}$] | | RMSE [mm day$^{-1}$] | | RE [%] | |
|---|---|---|---|---|---|---|
| | ARET | INCA | ARET | INCA | ARET | INCA |
| January | −0.01 | −0.05 | 0.29 | 0.34 | 1 | −7 |
| February | −0.17 | −0.30 | 0.60 | 0.65 | −12 | −25 |
| March | 0.04 | −0.23 | 0.84 | 0.89 | 4 | −14 |
| April | 0.80 | 0.66 | 1.34 | 1.59 | 35 | 28 |
| May | 0.79 | 0.51 | 1.38 | 1.58 | 29 | 19 |
| June | 0.19 | −0.24 | 1.42 | 1.80 | 6 | −8 |
| July | 0.39 | 0.31 | 1.29 | 1.58 | 12 | 9 |
| August | −0.09 | −0.01 | 1.16 | 1.42 | −1 | 1 |
| September | −0.14 | −0.10 | 0.96 | 1.11 | −6 | −4 |
| October | −0.15 | −0.06 | 0.57 | 0.69 | −8 | −3 |
| November | −0.03 | 0.01 | 0.43 | 0.54 | 2 | 5 |
| December | −0.16 | −0.18 | 0.39 | 0.43 | −15 | −18 |
| Year | 0.12 | 0.03 | 0.89 | 1.05 | 4 | −1 |

Another key question on which I have some concern is how AED based on potential ET makes sense in an environment where real ET could be very different from FAO conditions, having very different land covers such as forests, rocks, and so on.

This is indeed a very good question. However, we think it is far beyond the scope of the paper. It would be quite interesting to see these questions tackled in a separate paper.

L158 "Herein, SG-CL is driven with gridded observations and the OEKS15 dataset for the reference and future projection runs, respectively." The statement is not clear to me.

We will rephrase this sentence to:

"Herein, SG-CL is driven by gridded observations and the historical simulations of OEKS15 for the reference period and with scenario simulations of OEKS15 considering near and far future time periods."

**"glacier runoff"** This is also a key component of the CWB. The approach used is also not well explained. How a change in glacier area was estimated? How all the area above 2500 m is considered? Why this 2500 threshold?

Thanks for this important point.

We expand on the points raised referring to key literature

The model GLOGEM computes glacier mass balance and associated geometry changes for each glacier individually as described comprehensive in (Huss and Hock 2015) and (Huss and Hock 2018). The climatic mass balance is calculated at a monthly resolution based on near-surface air temperature and precipitation time series. Total mass changes are used to adjust each glacier's surface elevation and extent on a yearly basis using an empirical parameterization (Huss et al. 2010). 2500 m a.s.l. is used as a threshold for areas potentially impacted by glaciers as this is approximately the elevation above which glaciers can occur in the study area (e.g., Fischer et al. 2015).

Fischer A, Seiser B, Stocker Waldhuber M, et al (2015) Tracing glacier changes in Austria from the Little Ice Age to the present using a lidar-based high-resolution glacier inventory in Austria. Cryosph 9:753–766. doi: 10.5194/tc-9-753-2015

Huss M, Hock R (2015) A new model for global glacier change and sea-level rise. Front Earth Sci 3:1–22. doi: 10.3389/feart.2015.00054

Huss M, Hock R (2018) Global-scale hydrological response to future glacier mass loss. Nat Clim Chang 8:. doi: 10.1038/s41558-017-0049-x

Huss M, Jouvet G, Farinotti D, Bauder A (2010) Future high-mountain hydrology: A new parameterization of glacier retreat. Hydrol Earth Syst Sci 14:815–829. doi: 10.5194/hess-14-815-2010

"**3.5.2 Frequency Analysis - Return Periods**" Also here some parts are not clear.

How a return period can be calculated with not-stationary data?

Thank you for your important comment!

Of course, non-stationarity is an issue with respect to estimating return periods of a given variable. However, we tested the CWB for trends in the observational data, which revealed no significant trends in each season and class of altitude. This is a rather robust signal, since inter-annual variability of the CWB is known to be rather large. Given the short time periods of 30 years, insignificance of trends is rather plausible. This enables to assume that the data is stationary given no significant linear trends and justifies the application of the return period analysis.

We did the same for the climate scenario data, where we tested each of 30-year periods under consideration (3 periods, 2 classes of altitude, 4 seasons and 24 climate model runs), where we found only 12% of the total number of 30-year periods showed significant trends in the CWB. Given the large uncertainties already present in the climate scenarios, we argue that this minor fraction of non-stationary time periods would not alter the general conclusions of the return period estimates.

However, we will discuss this topic both in the methods and in the discussion section, because it is a rather relevant aspect of the applied methods.

10-y tr is not a too-low return period for an extreme drought?   For floods, usually, much larger Tr are considered.

Yes, that´s true. However, we thought of also displaying a more moderate drought event threshold. Since there is a more in-depth analysis of the 2003 event we think it would be worthwhile to keep this information on moderate drought changes.

**4 Results**:

Results are very interesting, especially for the spatial detail, which allows us to identify the local impacts of Alpine orography on the water budget. I am looking at what the 2022 summer drought will look like in terms of return periods in comparison with the 2003 one!

Yes, this would be really interesting, perhaps a topic for another paper with a wider historical context (HISTALP Data).

**Fig 3** I am surprised by the strong increase of precipitation of the RCP 4.5 scenario for 2070 – 2100.  Since most of the results are depending on this, and we know that climate models are quite uncertain in predicting precipitation trends concerning temperature, a discussion on the reliability of this high-precipitation RCP scenario would be very helpful.

Yes, we will discuss this feature in more detail as also indicated at the beginning.

**Fig 4d and L357.** I am surprised that the annual average of mountain regions is only 34 mm/year. Given the total precipitation that should exceed 1000 mm/year and ET that should not exceed 500 mm/year, 34 mm seems a very little number since mountains act as "water towers.

The altitudinal range of the mountainous areas are from 700-2500 m a.s.l., which rather wide. So, there are high proportions of areas near this lower bound, exhibiting higher values of AED. When averaging over the whole area, this might disturb the signal and perception that the mountainous areas should indicate higher values of the CWB.

**Fig 4e.** It is very interesting the increase of uncertainty in spring with the farthermost scenario. It is due to the major role of liquid precipitation? Please comment.

Yes, we will have a close look into this feature. Most probably, it is the uncertainty in the liquid precipitation and the antecedent snow pack conditions and subsequent snow melt.

**Fig 4f and L 292.** What do you mean by rainfall? Only liquid precipitation? It is not clear!

Thank you, we will specify in more detail. We will add the definition of terms in the methods section. Here, rainfall is meant as liquid precipitation.

**Fig 5.** Really beautiful Figure where the effect of elevation on the P/T relevance on changes in CWB is very clear. It would be nice to see the same Figure for different P and T scenarios may be in the supplementary material. What would happen without such a strong P increase?

**L371** Fig 6c not 7c

Thank you, we will correct that.

**L386** What is the drought duration? How do you define a drought period?

Here, a drought event is defined as a period in time where the CWB deceeds a certain threshold in the given season. This means, there is a hard definition for the drought duration, 3 months, which arises from the seasonal considerations.

**Acknowledgments:** Is it possible to acknowledge in addition the ADO project?

Yes, of course, we will acknowledge the ADO project.

---

## Author Comment (AC2)

Dear Reviewer,

Thank you for your comments and suggestions. Our response is indicated below your comments in blue font.

Major revisions are anticipated before the manuscript is considered for publication in the journal NHESS:

- This paper investigates future changes of surface water availability in Austria. Usually, researchers may use the previous observed data to verify the mathematical model, and then apply the model to deduce the trend in the future. Nevertheless, I am sorry it is difficult for me to find the evidences to prove the result is reliable.

   Thank you for your comment. Using bias corrected regional climate model projections for assessing future climate conditions on a regional scale is an extensively used method and subject of thousands of studies. Of course, one could question the reliability of future climate projections, but again, there are immense efforts taken to assess the biases and errors in climate models to quantify the uncertainty of future projections.

   Questioning the "mathematical model", which we assume means the climate model projections, would imply to discard all studies where future climate projections are used, until the model is verified.

   To this end, we think that these climate projections are still the best tools at hand, though prone to biases, errors and uncertainties, just because there are no other tools available. Given the vast demand on information on future climate impacts, there seems to our understanding to be no other, more reliably method.

   Here are some specific publications on the datasets and methods used to provide these climate scenarios for Austria:

   On the evaluation of Regional Climate Models over the European domain:

   doi:10.5194/gmd-7-1297-2014

   On the evaluation and guidelines for using the OEKS15 data set:

   doi:10.1016/j.cliser.2020.100179

   On the downscaling/bias correction method:

   https://doi.org/10.5194/hess-21-2649-2017

   Considering the representation of trends of the CWB in observations and models, we calculated trends within the reference period. Both the observations and the median across the model ensemble indicate a slightly positive trend, with +6.0 and +2.4 mm/year (±3mm/year uncertainty range) respectively. We therefore assume, that the given model ensemble is able to reproduce the observed climate trends, although uncertainties still persist, but are thoroughly discussed throughout the paper.

- No quantitative result is found whether in the parts Abstract or Conclusions.

  Thank you for your comment, we will provide most important quantitative findings in the Abstract as well as in the Conclusion section. To be more specific, we will add the numbers of the main findings, e.g. increase in average CWB during future time periods on an annual basis, as well as future return periods of a 2003 event under different emission scenarios.

- The parts "2 Data" and "3 methods" may be merged.

  Thank you for your comment. However, we think that separated sections make it easier to follow the content, since the methods section has already a rather deep hierarchy with 3 levels. Merging Data and Methods would increase these to 4 and in our opinion would reduce readability. Of course we are open for an editorial advice to restructure the article

- Description of the method is too tedious. The part "3 methods" may be greatly simplified. Many discussions, e.g., lines 102-106 and lines 121-130, may be moved to the part "Introduction" or "Discussion".

  We see your point in suggesting putting these paragraphs in the introduction or discussion section since they give basic information on the topics. However, we think it would be hard to understand for the reader, if there is a rather detailed introduction to CWB or AED in the Introduction, were the general topic (drought and water availability) is presented.

  We will try to shorten these paragraphs and condense the information to the most important points.

- Language errors or problems exist through the manuscript. Too many explanations included in the brackets, e.g., lines 269-274, make the paper unsmooth. Some long sentence, e.g., lines 63-65, is not readable. The first of the word, Where, in line 118, may be lowercase.

  Thanks you for your comment, we will go through the manuscript and will try to streamline and shorten the text. Considering your specific points:

  Lines 269-274:

  Figure 4: Monthly climate change signal of the CWB for lowlands in the upper panel and mountainous areas in the lower panel; (a,d) observed average monthly CWB in the reference period 1981-2010, where the shading denotes for the spatial variability of the CWB climatology, (b, e) ensemble median monthly climate change signal of the CWB for RCP4.5 (blue) and RCP2.6 (turquoise) for the near future in bold colour and the far future in pale colour, the shading denotes for the ensemble spread given by the 10th and 90th, (c, f) ensemble median monthly climate change signal of the individual CWB terms; rainfall: blue, snow melt: magenta, AED: green.

  Lines 63-65:

Although the body of existing literature points towards changing future water availability in Austria a comprehensive synopsis of all relevant processes altering surface water availability  is not accomplished yet, or just for small spatial entities (Hanus et al., 2021).

- Some tables, e.g., Table 2 in page 9, are not easy to understand.

  Yes, we try to make the table more easy to read.

---

## Author Response (AR1)

Dear Reviewer#1,

Thank you for your comments and suggestions. Our response is indicated below your comments in blue font.

Apart from that two main adaption have been applied to the paper:

1) We found an error in the estimation of Atmospheric Evaporative Demand (AED), since the longwave radiation was not considered in the first version. We did correct for that and re-derived the analysis. The results on the climate change signal with respect to AED are not affected, however AED in absolute terms is lower now, which affects the representation of the Climatic Water Balance in Figures 1, 4, 6, and 8 as well as Table 2.
We added following text to the Methods section to account for the correction:
*"However, calculating the reference data using station time-series, only shortwave net radiation was considered. Omitting the mainly outgoing longwave radiation leads to an overestimation of available energy on the surface and thus, an overestimation of potential evapotranspiration. To account for this incorrect representation of the energy balance in the initial ARET dataset, correction fields were applied. These were derived as the expected value (median per day of the year) of daily differences from 2013 to 2021 to Penman-Monteith reference evapotranspiration fields based on INCA input fields (Haiden et al. 2011), also considering outgoing longwave radiation."*

2) For the assessment of the future return period of a 2003-event we erroneously compared the observed CWB value of 2003 with the distribution of the projection. This analysis would be a mixture of real and model world. Instead we used the return period from the observational record and assessed the CWB of this respective return period in the reference period of the climate simulations. This CWB value serves as a reference for assessing the future return period in the scenario simulations of a similar event.
This affects first and foremost the results in section *3.2 Future change in extreme drought event probabilities.* However, the main conclusions are not affected by this correction.

**General comments**

This is valuable research, where changes in the climatic water budget (CWB) in the Alps, and future water scarcity risks are analyzed at an unprecedented spatial detail (1 km with a very good representation of mountain orographic factors).

It is also a work of great topical relevance and very timely, given the current 2022 heat wave in Europe and the water scarcity in the Alps.

This work explains well current and future alpine drought development mechanisms and shows how such events will become more likely in the future, despite possible year-round greater water availability in the Alps. Positive CWB signals are mainly caused by increasing rainfall, particularly during winter. On the contrary, the negative CWB changes during summer are caused by both increasing AED and slightly decreasing rainfall. Because of less snowpack contribution, water provision in the summer will become less constant and more connected to the liquid precipitation variability, and therefore, given the higher temperatures, summer water scarcity is more likely.

The figures are excellent ad full of information. Results are very interesting, especially for the spatial detail, which allows us to identify the local impacts of Alpine orography on the water budget. One relevant result is the finding that in the far future period the CC signal is much stronger in mountains than in lowlands.

I am looking for further research, showing how the 2022 summer drought will be compared to 2003 in terms of return period!

However, several methodological details need further clarification, especially on evaporation and glacier runoff estimation, and should be addressed before publication, as indicated below and in the specific comments.

- I am surprised by the strong increase of precipitation of the RCP 4.5 scenario for 2070 – 2100. Since most of the results are depending on this, and we know that climate models are still more uncertain in predicting precipitation trends than temperature, a discussion on the reliability of this high-precipitation RCP scenario would be very helpful.

  Yes, you are right, and thank you for your important comment, precipitation projections are indeed highly uncertain, particularly in the Alpine region due to the complex generation of precipitation and the high fraction of convective precipitation. However, it is a feature seen also in other publications, eg. https://doi.org/10.1002/2017JD027176. Here, an increase in precipitation in winter (+17mm/d), spring and autumn (+4mm/d) is contrasted by a decrease in summer (-4mm/d), for RCP4.5, far future (2070-2099), which in sum points towards more precipitation on an annual basis as also indicated in the present paper for Austria.

  We added following paragraph to the Discussion section:

  *"This increase in precipitation is not only a feature in this study using this specific model ensemble, but also in several others (Kotlarski et al., 2023). The origins of this trend towards more wetness in winter and spring is not fully understood. However, Rajczak and Schär (2017) point towards the specific location of the alpine region in a transition zone between future precipitation increase (decrease) on the northern (southern) side of the main alpine crest."*

- I am also a little surprised you do not find an important increase in Evaporative Demand. I am just wondering if this could be related to the quite simplified approach used to estimate evapotranspiration ET. This point deserves further discussion.

  Yes, we will further discuss this crucial aspect of the paper in more detail. See particular suggestions below.

- **Title and journal.** The title triggers curiosity, but then reading the paper I found much more than this. The best parts of the paper are the high-resolution scenarios of climatic water balance (CWB) for Austria. So, the title is a little bit limiting. Changes in drought risk are only one of the aspects. I am also wondering if this paper fits better with HESS instead of NHESS.

  Thank you for your kind comment on the paper. You are right, there is more in it than just the drought aspect. However, we think that all the other topics rotate around the

drought topic, which is from our perspective the most important point of the paper. This is the reason we choose to publish in NHESS, whereas HESS would also make perfectly sense. We don´t know if a change in journal would be easily to accomplish. If things getting too tricky we would stick to NHESS.

Rethinking the title we changed it to:

**Specific comments**

**Abstract.** Please specify that you consider only meteorological drought.

Yes, good point. We specified that we are explicitly considering meteorological drought:

*"Apart from analysing the mean changes of these components we also pursue a hazard risk approach by estimating future changes in return periods of meteorological drought events of a given magnitude as observed in the reference period."*

1. **Introduction (and title).**

L68 "How will future surface water availability change" I am not sure if the term surface water availability is correct. You are working with RCMS and a very simplified hydrological representation of the processes. What you get is mostly the water budget. Then you work on the CWB, which is simply a good indicator of the real surface water availability. Why do not also refer to CWB in the title?

Ok, thank you for your comment. It is true, that we do not fully represent the surface water availability, so referring to the CWB in the title would be more appropriate, see new title above.

To have an estimation of surface water you need to model also water infiltration, vadose zone hydrology, and groundwater recharge losses… which is of course beyond the purpose of the paper.

**2 Data**:

L80 "The broadly used RCP8.5 scenario is intentionally not included here, since its emission pathway is highly unlikely from today´s emissions trajectories,"

Sure? Are you so optimistic about the future?

The literature on criticism considering overemphasising the RCP8.5 scenario is constantly growing. To make the results not too overloaded we decided to display RCP4.5 and RCP2.6 as one pathway which is optimistic and plausible and another one that is more optimistic. We will therefore better argue on limiting to those two, also highlighting the latest literature on RCP scenarios. Therefore we added new literature to underpin our statement:

*"From today's perspective an emission path following RCP4.5 is, at least until 2030, the most likely one given current estimates (UNFCCC, 2022)."*

We also added a paragraph to the discussion section were we discuss potential feedbacks that might introduce additional warming albeit emission are not increasing:

*"We intentionally do not use an extreme emission scenario like RCP8.5, as the literature is growing on the implausible exaggeration of future emissions of this socio-economic pathway (see details in the data section). Although RCP4.5 might be seen as the most plausible scenario at current times, even larger global temperature increases cannot be completely discarded. Non-linear behavior of the climate system and the potential of crossing tipping points may cause additional warming, leading to temperature levels far beyond current plausible scenarios like RCP4.5. Although the existence of some tipping points and further the timing of them in general is still under debate (Brovkin et al., 2021; Lenton et al., 2008). Uncertainties in the climate system arising from this matter have to be kept in mind."*

L87 "EURO-CORDEX downscaling," Could you provide more information on the approach? How is orography taken into account for the downscaling?

Thanks for your comment, we will discuss the downscaling procedure in more detail. Orography is indirectly taken into account, since the reference data (SPARTACUS) does explicitly consider orographic effects (cold air pools, inversions etc.) see these two publications for details:

https://doi.org/10.1007/s00704-017-2093-x

https://doi.org/10.1007/s00704-015-1411-4

We added following text to the Data section:

*"Both reference datasets are considering orographic effects on temperature (e.g. cold air pool formation, foehn effects) and on precipitation (orographic precipitation) which is rather important for interpolating climatic variables in complex terrain of the Austrian domain. The basic data sets for OEKS15 (EURO-CORDEX) were thoroughly evaluated in Kotlarski et al. (2014) and OEKS15 was evaluated and a comprehensive guide line given on the usage in Chimani et al. (2020)."*

**3 Methods**:

L119 "snowmelt model," Ok it is essential to consider snow melt in the CWB, but then you need a good snow melt model. Please provide here or later more information on how snow melt is modeled.

Yes, good point, we added this rather relevant information in the methods section:

*"The latter is calculated from clear-sky solar radiation model output and cloudiness derived from daily temperature amplitudes as well as surface albedo (weighted average of snow) and snow-free albedo using CORINE land cover types and albedo values from the literature. The actual incoming shortwave radiation is computed as a product of clear-sky incoming shortwave radiation (Reference IqbalIqbal, 1983; Reference CorripioCorripio, 2003) and a cloud transmission factor, representing the attenuation of solar radiation by clouds. The*

*clear-sky incoming shortwave radiation is calculated as the sum of direct, diffuse and reflected shortwave radiation and requires knowledge of the exact position of the Sun and its interaction with the surface topography, as well as the transmissivity of the atmosphere."*

"**3.2 Atmospheric Evaporative Demand"** AED is a key parameter. For this is essential the calculation of ETP. Which is the accuracy of the calculated ETP? How has been validated?

Yes, AED calculated by the given formulation of a re-calibrated Hargreaves equation and is evaluated in the referred publication:

https://doi.org/10.5194/hess-20-1211-2016

In comparison with station data Penman-Monteith (nearest neighbour gridpoint) and a Penman-Monteith estimate using data from the Austrian nowcasting system INCA the given method (ARET) is performing rather well. However, during the analysis we found out that INCA is not considering longwave radiation, which made a correction with a monthly correction factor necessary. See text at the very beginning of the document.

Apart from that we did an even more comprehensive comparison of trends and sensitivities auf AED trends in observations with the re-calibrated Hargreaves and Penman-Monteith methods and also compared to climate simulations. Following text and analysis is added:

*"We furthermore assessed the relationship between changes in AED and temperature, applied both for the observational and scenario data. The temperature trend over the entire Austrian domain from 1977-2014 is +0.47 °C decade-1 (SPARTACUS data), which relates to an AED trend of 17.2 mm year-1 decade-1 (see above). This yields an AED increase of +36.6 mm year-1 °C-1. For the climate scenarios, based exemplarily on RCP4.5, from 2010-2050 a temperature increase of +0.28 °C decade-1 is apparent, compared to an AED increase of +10.1 mm year-1 decade-1. These results indicate a scaling of +36.1 mm year-1 °C-1 of AED with a given temperature forcing, which is in very close agreement with the observed value of 36.6 mm year-1 decade-1. These results of the temperature scaling and the good agreement of the observed trends between AED of Duethmann and Blöschl (2018) and the one following the approach of Haslinger and Bartsch (2016) using a re-calibrated Hargreaves formulation proves that this simpler AED method is able to provide a physically sound representation of the main processes driving changes in AED."*

Another key question on which I have some concern is how AED based on potential ET makes sense in an environment where real ET could be very different from FAO conditions, having very different land covers such as forests, rocks, and so on.

This is indeed a very good question. However, we think it is far beyond the scope of the paper. It would be quite interesting to see these questions tackled in a separate paper.

L158 "Herein, SG-CL is driven with gridded observations and the OEKS15 dataset for the reference and future projection runs, respectively." The statement is not clear to me.

This sentence is corrected to:

"Herein, SG-CL is driven by gridded observations and the historical simulations of OEKS15 for the reference period and with scenario simulations of OEKS15 considering near and far future time periods."

"**glacier runoff**" This is also a key component of the CWB. The approach used is also not well explained. How a change in glacier area was estimated? How all the area above 2500 m is considered? Why this 2500 threshold?

We added the following information to this section:

*"GLOGEM computes glacier mass balance and associated geometry changes for each glacier individually as described comprehensive in Huss and Hock (2015) and Huss and Hock (2018). The climatic mass balance is calculated at a monthly resolution based on near-surface air temperature and precipitation time series. Total mass changes are used to adjust each glacier's surface elevation and extent on a yearly basis using an empirical parameterization (Huss et al., 2010)."*

And

*"2500 m a.s.l. is used as a threshold for areas potentially impacted by glaciers as this is approximately the elevation above which glaciers can occur in the study area (Fischer et al., 2015)."*

"**3.5.2 Frequency Analysis - Return Periods**" Also here some parts are not clear.

How a return period can be calculated with not-stationary data?

Non-stationarity is indeed an issue when assessing return periods. We test for non-stationarity and assessed it for observations and climate simulations. We added following analysis and text to the Methods section:

*"Similar results are obtained assessing the stationarity of the different 30-year time periods considered, which is a general assumption of classical frequency analysis (Coles, 2001). We tested the observations as well as the climate scenario time periods for significant trends as an indication for non-stationarity. In the observations all 30-year time periods investigated (for each season and for lowland and mountainous areas) do not show significant trends of the CWB following the Mann-Kendall trend test. These results are in line with similar investigations by Blöschl et al. (2018)who could show that increasing AED is balanced by an increase in precipitation. Considering the climate simulations, 13% of all individual 30-year periods (576 in total given by 24 individual runs times 4 seasons times 3 time periods times 2 different areas) show significant trends at the 5% level, thereby indicating non-stationarity. However, since this is only a minor fraction, and 30-year time-slices are relatively short for assessing the stationarity of climate simulations, we consider classical extreme value theory to be generally applicable, while the related uncertainties are taken into account when interpreting the results."*

10-y tr is not a too-low return period for an extreme drought? For floods, usually, much larger Tr are considered.

Yes, that´s true. However, we thought of also displaying a more moderate drought event threshold. Since there is a more in-depth analysis of the 2003 event we think it would be worthwhile to keep this information on moderate drought changes.

**4 Results**:

Results are very interesting, especially for the spatial detail, which allows us to identify the local impacts of Alpine orography on the water budget. I am looking at what the 2022 summer drought will look like in terms of return periods in comparison with the 2003 one!

Yes, this would be really interesting, perhaps a topic for another paper with a wider historical context (HISTALP Data).

**Fig 3** I am surprised by the strong increase of precipitation of the RCP 4.5 scenario for 2070 – 2100. Since most of the results are depending on this, and we know that climate models are quite uncertain in predicting precipitation trends concerning temperature, a discussion on the reliability of this high-precipitation RCP scenario would be very helpful.

Yes, we will discuss this feature in more detail as also indicated at the beginning. As indicated above we added following text to the Discussion section:

*"This increase in precipitation is not only a feature in this study using this specific model ensemble, but also in several others (Kotlarski et al., 2023). The origins of this trend towards more wetness in winter and spring is not fully understood. However, Rajczak and Schär (2017) point towards the specific location of the alpine region in a transition zone between future precipitation increase (decrease) on the northern (southern) side of the main alpine crest."*

**Fig 4d and L357.** I am surprised that the annual average of mountain regions is only 34 mm/year. Given the total precipitation that should exceed 1000 mm/year and ET that should not exceed 500 mm/year, 34 mm seems a very little number since mountains act as "water towers.

You are right, these are erroneous values. Two errors occurred, one is the biased assessment of AED, which we corrected for, and the other is that for this number the monthly values are averaged instead of summed. Now plausible values for all levels of altitude are achieved.

**Fig 4e.** It is very interesting the increase of uncertainty in spring with the farthermost scenario. It is due to the major role of liquid precipitation? Please comment.

It is indeed the uncertainty in the liquid precipitation and the antecedent snow pack conditions and subsequent snow melt. However, a detailed analysis of this topic is beyond the scope of the current paper, but would be a rather interesting research question for future activities.

**Fig 4f and L 292.** What do you mean by rainfall? Only liquid precipitation? It is not clear!

Thank you, we specified that the terms liquid precipitation and rainfall are meaning the same.

**Fig 5.** Really beautiful Figure where the effect of elevation on the P/T relevance on changes in CWB is very clear. It would be nice to see the same Figure for different P and T scenarios may be in the supplementary material. What would happen without such a strong P increase?

**L371** Fig 6c not 7c

Thank you, corrected.

**L386** What is the drought duration? How do you define a drought period?

Here, a drought event is defined as the deceedance of the CWB of a certain threshold in the given season. So there is a hard definition for the drought duration, 3 months, which arises from the seasonal considerations.

**Acknowledgments:** Is it possible to acknowledge in addition the ADO project?

Yes, of course, we will acknowledge the ADO project.

Dear Reviewer#2,

Thank you for your comments and suggestions. Our response is indicated below your comments in blue font.

Apart from that two main adaption have been applied to the paper:

3) We found an error in the estimation of Atmospheric Evaporative Demand (AED), since the longwave radiation was not considered in the first version. We did correct for that and re-derived the analysis. The results on the climate change signal with respect to AED are not affected, however AED in absolute terms is lower now, which affects the representation of the Climatic Water Balance in Figures 1, 4, 6, and 8 as well as Table 2.
   We added following text to the Methods section to account for the correction:
   *"However, calculating the reference data using station time-series, only shortwave net radiation was considered. Omitting the mainly outgoing longwave radiation leads to an overestimation of available energy on the surface and thus, an overestimation of potential evapotranspiration. To account for this incorrect representation of the energy balance in the initial ARET dataset, correction fields were applied. These were derived as the expected value (median per day of the year) of daily differences from 2013 to 2021 to Penman-Monteith reference evapotranspiration fields based on INCA input fields (Haiden et al. 2011), also considering outgoing longwave radiation."*

4) For the assessment of the future return period of a 2003-event we erroneously compared the observed CWB value of 2003 with the distribution of the projection. This analysis would be a mixture of real and model world. Instead we used the return period from the observational record and assessed the CWB of this respective return period in the reference period of the climate simulations. This CWB value serves as a reference for assessing the future return period in the scenario simulations of a similar event.
   This affects first and foremost the results in section *3.2 Future change in extreme drought event probabilities.* However, the main conclusions are not affected by this correction.

Major revisions are anticipated before the manuscript is considered for publication in the journal NHESS:

- This paper investigates future changes of surface water availability in Austria. Usually, researchers may use the previous observed data to verify the mathematical model, and then apply the model to deduce the trend in the future. Nevertheless, I am sorry it is difficult for me to find the evidences to prove the result is reliable.

  Thank you for your comment. Using bias corrected regional climate model projections for assessing future climate conditions on a regional scale is an extensively used method and subject of thousands of studies. Of course, one could question the reliability of future climate projections, but again, there are immense efforts taken to assess the biases and errors in climate models to quantify the uncertainty of future projections.

  Questioning the "mathematical model", which we assume means the climate model projections, would imply to discard all studies where future climate projections are used, until the model is verified.

To this end, we think that these climate projections are still the best tools at hand, though prone to biases, errors and uncertainties, just because there are no other tools available. Given the vast demand on information on future climate impacts, there seems to our understanding no other, more reliably method.

Here are some specific publications on the datasets and methods used to provide these climate scenarios for Austria:

On the evaluation of Regional Climate Models over the European domain:

doi:10.5194/gmd-7-1297-2014

On the evaluation and guidelines for using the OEKS15 data set:

doi:10.1016/j.cliser.2020.100179

On the downscaling/bias correction method:

https://doi.org/10.5194/hess-21-2649-2017

We also added the following text to the Data section:

*"The basic data sets for OEKS15 (EURO-CORDEX) were thoroughly evaluated in Kotlarski et al. (2014) and OEKS15 was evaluated and a comprehensive guide line given on the usage in Chimani et al. (2020)."*

- No quantitative result is found whether in the parts Abstract or Conclusions.

  Thank you for your comment, we provide most important quantitative findings in the Abstract as well as in the Conclusion section:

  *"The results show in general wetter conditions over the course of the 21st century over Austria on an annual basis compared to the reference period 1981-2010 (e.g. RCP4.5 +107 mm, RCP2.6 +63 mm for the period 2071-2100). Considering seasonal differences, winter and spring are getting wetter due to an increase in precipitation and a higher fraction of rainfall as a consequence of rising temperatures. In summer only little changes in the mean of the climatic water balance conditions are visible across the model ensemble (e.g. RCP4.5 ±0mm, RCP2.6 -2 mm for the period 2071-2100)."*

  *"In this study we presented scenarios of future surface water availability across different elevations and for the most important variables of the CWB. In general, wetter conditions are projected on an annual basis (e.g. RCP4.5 +107 mm, RCP2.6 +63 mm for the period 2071-2100), however particular seasonal shifts of snow melt and glacier melt at higher elevations change the surface water availability in the course of the year. Drought conditions in particular are expected to become more frequent during summer months, which is predominantly driven by an increase in*

*interannual variability of the CWB. For a 2003-event like situation during summer in the lowlands, the return period is estimated to decrease from 36 years to 20 years under RCP4.5 in the near future for example."*

- The parts "2 Data" and "3 methods" may be merged.

  Thank you for your comment. However, we think that separated sections are better guiding the reader, since the methods section has already a rather deep hierarchy with 3 levels. Merging Data and Methods would increase these to 4. Journal guidelines often recommend not to use too many subsections.

- Description of the method is too tedious. The part "3 methods" may be greatly simplified. Many discussions, e.g., lines 102-106 and lines 121-130, may be moved to the part "Introduction" or "Discussion".

  We see your point in suggesting putting these paragraphs in the introduction or discussion section since they give basic information on the topics. However, we think it would be hard to understand for the reader, if there is a rather detailed introduction to CWB or AED in the Introduction, were the general topic (drought and water availability) is presented.

- Language errors or problems exist through the manuscript. Too many explanations included in the brackets, e.g., lines 269-274, make the paper unsmooth. Some long sentence, e.g., lines 63-65, is not readable. The first of the word, Where, in line 118, may be lowercase.

  Thanks you for your comment, we went through the manuscript and tried to streamline the text on various sections.

- Some tables, e.g., Table 2 in page 9, are not easy to understand.

  Yes, thanks for your comment, we redesigned the table and think it increases the readability essentially.

---

## Author Response (AR2)

Referee #1

**Suggestions for revision or reasons for rejection**

Please revise again figures and numbers accurately.

Major errors and inconsistencies have been corrected.

Now yearly water budget is more reasonable in mountain regions.

Thank you for your positive response. We revised the figure numbering throughout the manuscript accordingly and corrected apparent errors.

Referee #2

**Suggestions for revision or reasons for rejection**

Major revision

I am sending my understanding for the rejection of most of my comments because I also think maybe the authors can enlighten readers using their own ways.

Thank you for your comment.

Nevertheless, undisputed language errors exist throughout the manuscript. For examples: (1) In page 3, the general title of Figure 3 is needed, and the map on the top-left corner of Figure 1a is not clear. (2) I am lost that the third section of the manuscript is the method, "3 Methods" (page 4), and it is also the result, "3 Results" (page 10).

Thank you for your comments, however I´m not quite sure to understand them all correctly. The Figure on page 3 is figure 1. The respective caption can be found beneath the figure, a title for each figure is not mandatory following the journal layout rules. The figure in the top-left corner of 1a should just indicate the location of the study region in a broader for readers not generally familiar with the geographical location of Austria. It should just be a guidance for the reader. However, we added to the figure caption: …" (a) Topography of Austria, the inset figure indicates the location on the European domain,…"

You are right, Methods and Results are number both as 3, Results should be 4, which is corrected now. Thanks for pointing towards this error.

In addition, although the authors thought "I am sorry it is difficult for me to find the evidences to prove the result is reliable" (in page 9 of the author response), the authors may try to publish the paper after they have gotten the evidences.

Thank you for your comment.